# LSD1/PRMT6-targeting gene therapy to attenuate androgen receptor toxic gain-of-function ameliorates spinobulbar muscular atrophy phenotypes in flies and mice

Ramachandran Prakasam[1,17], Angela Bonadiman [2,17], Roberta Andreotti[3,4,5,17], Emanuela Zuccaro[3,4,5], Davide Dalfovo [2], Caterina Marchioretti[3,4,5], Debasmita Tripathy[2], Gianluca Petris [2,15,16], Eric N. Anderson[6], Alice Migazzi [1,2], Laura Tosatto [1], Anna Cereseto [2], Elena Battaglioli[7], Gianni Sorarù[5,8], Wooi Fang Lim [9,10], Carlo Rinaldi[9,10], Fabio Sambataro[5,8], Naemeh Pourshafie[11], Christopher Grunseich[11], Alessandro Romanel [2], Udai Bhan Pandey [6], Andrea Contestabile [12], Giuseppe Ronzitti [13,14], Manuela Basso [2] ✉ & Maria Pennuto [1,3,4,5] ✉

Spinobulbar muscular atrophy (SBMA) is caused by CAG expansions in the androgen receptor gene. Androgen binding to polyQ-expanded androgen receptor triggers SBMA through a combination of toxic gain-of-function and loss-of-function mechanisms. Leveraging cell lines, mice, and patient-derived specimens, we show that androgen receptor co-regulators lysine-specific demethylase 1 (LSD1) and protein arginine methyltransferase 6 (PRMT6) are overexpressed in an androgen-dependent manner specifically in the skeletal muscle of SBMA patients and mice. LSD1 and PRMT6 cooperatively and synergistically transactivate androgen receptor, and their effect is enhanced by expanded polyQ. Pharmacological and genetic silencing of LSD1 and PRMT6 attenuates polyQ-expanded androgen receptor transactivation in SBMA cells and suppresses toxicity in SBMA flies, and a preclinical approach based on miRNA-mediated silencing of LSD1 and PRMT6 attenuates disease manifestations in SBMA mice. These observations suggest that targeting overexpressed co-regulators can attenuate androgen receptor toxic gain-of-function without exacerbating loss-of-function, highlighting a potential therapeutic strategy for patients with SBMA.

Spinobulbar muscular atrophy (SBMA), also known as Kennedy's disease, is an X-linked late-onset neuromuscular disease that affects 2–5 in 100,000 people worldwide. No therapy exists to cure SBMA or delay disease onset and progression. SBMA is caused by microsatellite expansions (≥38 repeats) of a glutamine (Q)-encoding CAG triplet tandem repeat in exon 1 of the androgen receptor (*AR*) gene,

resulting in production of an AR with an aberrantly elongated polyQ tract[1]. SBMA belongs to the family of diseases caused by polyglutamine (polyQ) expansions, which include Huntington's disease, dentatorubral-pallidoluysian atrophy, and six types of spinocerebellar ataxia[2]. SBMA is characterized by selective degeneration of lower motor neurons[3,4], yet emerging research also demonstrates

primary involvement of peripheral tissues such as skeletal muscle[5]. Phenotypes are attributed largely to toxic gain-of-function (GOF) of polyQ-expanded AR[5]. However, mild signs of androgen insensitivity and endocrine abnormalities also implicate partial AR loss-of-function (LOF) in SBMA[6,7]. A necessary step for toxicity is binding of AR with its natural ligands, testosterone, and its more potent derivative dihydrotestosterone (DHT)[8]. As such, SBMA is unique among polyQ diseases for its sex specificity (in humans as well as in fly and mouse models)[8–11]—males develop severe symptoms, while females develop mild or no symptoms even if homozygous for the mutation[12]. Although the androgen-dependent nature of disease and experimental evidence support chemical castration as a therapeutic strategy for SBMA, clinical trials based on this approach show benefits only in a subset of patients[13]. Moreover, chronic androgen ablation may enhance symptoms associated with androgen deprivation, spanning from muscle atrophy and weakness to metabolic alterations and depression. Thus, alternative strategies are necessary to improve clinical outcomes.

AR is a transcription factor activated by androgens. PolyQ expansions alter the native function of AR, resulting in aberrant gene expression in both motor neurons and muscle cells[14–16]. To properly exert its pleiotropic functions in different tissues, AR interacts with transcriptional co-regulators. This interaction is necessary for neurodegeneration in flies[17], and pharmacological strategies to prevent interaction of AR with co-regulators can attenuate the phenotype of transgenic SBMA mice[18,19]. However, this approach may enhance the LOF of AR signaling, thus exacerbating androgen-insensitivity symptoms recurrent in SBMA patients. A potential alternative strategy to attenuate the toxic GOF of polyQ-expanded AR without enhancing LOF may be to target native co-regulators aberrantly overexpressed or overactivated by dysfunctional androgen signaling. Identification of structural and functional interactions of polyQ-expanded AR with co-regulators is thus necessary to inform appropriate targets for therapy development.

A valuable example of how polyQ-expansions can alter the interaction of AR with its co-regulators is represented by protein arginine methyltransferase 6 (PRMT6)[20]. PRMT6 is a general transcriptional co-repressor yet also is a co-activator of AR[21]. PRMT6 expression is often upregulated in hormone-dependent cancers, such as prostate cancer[22,23], and in mouse models of metabolic syndrome, diabetes, and insulin resistance[24,25]—symptoms that also are present in ~50% of SBMA patients[5]. Previous transcriptomic analysis identified significant upregulation of *Prmt6* in the skeletal muscle of a mouse model of SBMA[16]. Similar to PRMT6, lysine-specific demethylase 1 (LSD1, AOF2, or KDM1A) is a transcriptional co-activator of AR[26] and is upregulated in prostate cancer[27]. Notably, both LSD1 and PRMT6 have a steroid hormone binding motif, LXXLL (where L is leucine and X is any amino acid). Reasoning by analogy, here we addressed whether LSD1 trans-activates polyQ-expanded AR and explored whether there is any molecular link between LSD1 and PRMT6 underlying disease pathogenesis. We obtained evidence that suppression of AR co-regulators, which are overexpressed in SBMA skeletal muscle and that synergistically enhance mutant AR toxic GOF, could be an effective therapeutic strategy for SBMA.

## Results
### LSD1 and PRMT6 are induced by androgens in SBMA skeletal muscle
We asked whether LSD1 and PRMT6 are aberrantly expressed in tissues that degenerate in SBMA, including skeletal muscle, liver, spinal cord, brainstem, and heart[5]. To address this question, we used transgenic mice expressing human AR100Q that we recently generated and characterized[28,29], knock-in SBMA mice in which *AR* exon 1 was replaced with the human *AR* exon 1 coding for a polyQ-expanded AR with 113Q[10], and available patient biopsies (Supplementary Table 1). AR100Q male

mice are non-symptomatic at 4 weeks of age (pre-symptomatic stage), start to show signs of muscle atrophy and motor dysfunction by 8 weeks of age (onset), and manifest signs of denervation by 12 weeks of age (late stage). Real-time PCR analysis showed that *Lsd1* and *Prmt6* transcript levels were significantly increased two-fold in skeletal muscle (quadriceps) and liver of male SBMA mice at the pre-symptomatic stage compared to wildtype (WT) male mice (Fig. 1a). In the brainstem, spinal cord, and heart, transcript levels of *Lsd1* and *Prmt6* were significantly higher only in the brainstem at 4 weeks of age and to a lower extent than in skeletal muscle and liver. In muscle, upregulation was detected before the onset of motor dysfunction and denervation[28] and was constant and persistent at all disease stages (4–12 weeks). Western blotting confirmed LSD1 and PRMT6 upregulation at the protein level in SBMA quadriceps muscle, but not spinal cord, compared to controls (Fig. 1b, Supplementary Fig. 1a).

In muscle of female AR100Q mice, *Lsd1* and *Prmt6* transcript levels were upregulated only at late stage of disease and to a lower level (1.5-fold for *Lsd1* and 1.8-fold for *Prmt6*) than male mice (Fig. 1c). Further, *Lsd1* and *Prmt6* were significantly upregulated 5–10-fold in the fast-twitch extensor digitorum longus muscle of male AR100Q mice, and overexpression was normalized by surgical castration, consistent with the androgen-dependent nature of SBMA (Fig. 1d). In addition, *Lsd1* and *Prmt6* transcript levels were significantly upregulated 4-fold and 3-fold, respectively, in the skeletal muscle of AR113Q male knock-in mice compared to WT controls (Fig. 1e). Similar results were obtained in C2C12 myoblasts overexpressing AR100Q differentiated to myotubes, suggesting that this phenomenon occurs in muscle in a cell-autonomous fashion (Fig. 1f)[30,31]. Importantly, *LSD1* and *PRMT6* transcript levels also were upregulated 1.5-fold and 2.5-fold in the muscle of SBMA patients compared to controls, thus underlying the relevance of these findings to disease (Fig. 1g). *LSD1* and *PRMT6* overexpression was not detected in patient-derived induced pluripotent stem cells (iPSCs) differentiated to motor neurons, as well as patient liver biopsies, suggesting that similar to SBMA mice, overexpression of these AR co-regulators is specific to skeletal muscle also in patients (Supplementary Fig. 1b).

To determine whether polyQ-expanded AR directly affects *Lsd1* and *Prmt6* transcription, we searched for putative androgen-responsive elements (AREs) in enhancers and distal/core promoters of both genes. Bioinformatics analysis showed a putative ARE in the promoter of *Lsd1* and *Prmt6* (Fig. 1h). In a chromatin-immunoprecipitation (ChIP) assay of C2C12 myoblasts expressing AR with a normal (AR24Q) or pathogenic (AR100Q) polyQ tract[31,32], we detected increased occupancy of polyQ-expanded AR at the ARE sites of both *Lsd1* and *Prmt6* promoters (Fig. 1h, Supplementary Fig. 1c). These results provide a molecular mechanism underlying *Lsd1* and *Prmt6* overexpression in SBMA myofibers.

Taken together, these results indicate that *Lsd1* and *Prmt6* are overexpressed in a muscle cell-autonomous fashion, mainly as a result of an androgen-dependent AR toxic GOF.

### LSD1 is a co-activator of polyQ-expanded AR
Based on these observations, we further investigated the relationship of AR, PRMT6, and LSD1 in the context of SBMA pathogenesis. We previously showed that PRMT6 interacts with polyQ-expanded AR[20]. To address whether LSD1 forms a complex with polyQ-expanded AR, we expressed Flag-tagged AR24Q or AR65Q with or without LSD1 in human embryonic kidney 293T (HEK293T) cells and processed the cells for co-immunoprecipitation assays. Importantly, overexpression of LSD1 per se did not modify AR expression under these experimental conditions (Supplementary Fig. 2a). Pull-down of Flag-tagged normal and polyQ-expanded AR co-immunoprecipitated LSD1 (Supplementary Fig. 2a). In addition to LSD1, at least three more protein isoforms derived by alternative splicing (LSD1-2a, LSD-8a, and LSD1-2a/8a)[33] formed a complex with normal and polyQ-expanded AR both in the absence and presence of DHT (Supplementary Fig. 2b).

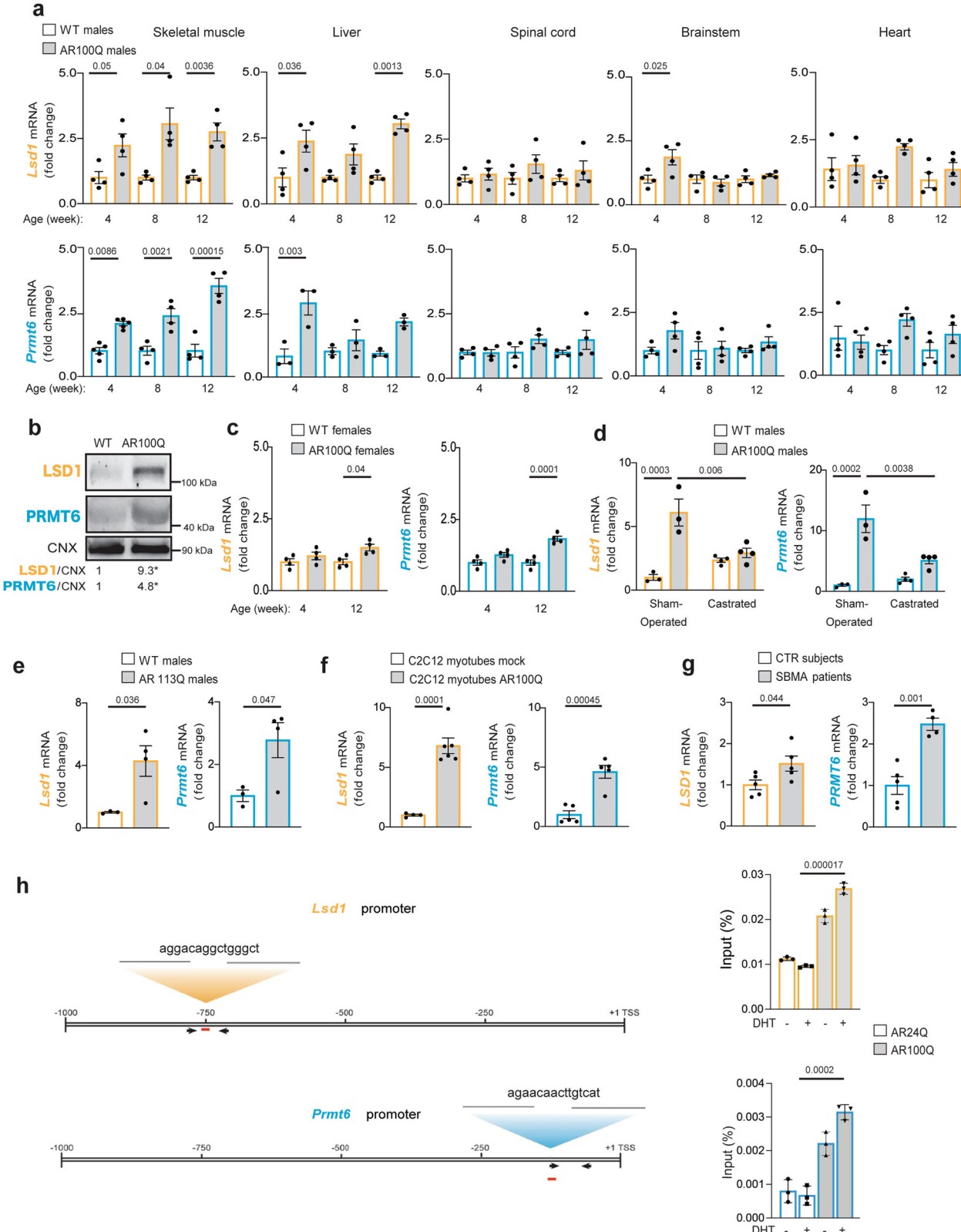

To examine the AR–LSD1 interaction in cells vulnerable in SBMA, we used motor neuron-derived MN1 cells and patient-derived cells[30]. Immunofluorescence analysis showed that normal and polyQ-expanded AR localized with endogenous LSD1 in MN1 cells treated with vehicle or DHT (Supplementary Fig. 3a), patient-derived iPSCs differentiated to motor neurons (Supplementary Fig. 3b), and the spinal cord of an SBMA patient (Supplementary Fig. 3c). Notably, AR,

LSD1, and PRMT6 were all present in the nuclei of human iPSC-derived motor neurons. To assess protein–protein interactions in the subcellular compartments of intact cells, we leveraged a proximity ligation assay (PLA), which is based on oligonucleotide-conjugated secondary antibodies detecting in situ protein–protein interactions. As a positive control, we analyzed the interaction of polyQ-expanded AR and PRMT6, which is enhanced in the presence of DHT, as shown in our

**Fig. 1 | Early and persistent androgen-dependent overexpression of LSD1 and PRMT6 in the skeletal muscle of SBMA mice and patients. a** RT-PCR analysis of *Lsd1* and *Prmt6* transcript levels in the indicated tissues of WT and AR100Q mice (mice/genotype/age: LSD1 *n* = 4; PRMT6: skeletal muscle *n* = 5 at 4 weeks, *n* = 4 at 8–12 weeks; liver: *n* = 3; heart, brainstem and spinal cord: *n* = 4). **b** Western blots of LSD1 and PRMT6 levels in the quadriceps muscle of 12-week-old WT and AR100Q mice (*n* = 4 mice/genotype). \**p* = 0.048 LSD1, and *p* = 0.023 PRMT6. LSD1 and PRMT6 were detected with specific antibodies, and calnexin (CNX) was used as loading control. **c** RT-PCR analysis of *Lsd1* and *Prmt6* transcript levels in the quadriceps muscle of female WT and AR100Q mice (*n* = 4 mice/genotype). **d** RT-PCR analysis of *Lsd1* and *Prmt6* transcript levels in the extensor digitorum longus muscle of sham-operated and surgically castrated 8-week-old male AR100Q mice (*n* = 3 WT and AR100Q sham-operated mice and *n* = 4 WT and AR100Q castrated mice). **e** RT-PCR analysis of *Lsd1* and *Prmt6* transcript levels in the quadriceps muscle of 24-week-old male WT and knock-in mice expressing AR113Q (*n* = 3 WT

and 4 knock-in mice). **f** RT-PCR analysis of *Lsd1* and *Prmt6* transcript levels in C2C12 myoblasts stably transduced with empty lentiviral vector (mock) or vector expressing AR100Q and differentiated to myotubes in presence of DHT (10 nM, 10 days, LSD1 expression n = 4 biological replicates mock, and *n* = 6 biological replicates AR100Q; PRMT6 *n* = 5 biological replicates). **g** RT-PCR analysis of *LSD1* and *PRMT6* transcript levels of quadriceps muscle biopsies of control (CTR) subjects and SBMA patients (*n* = 5 participants/genotype). **h** (Left) Schematic of putative androgen-responsive elements in the promoter of Lsd1 and Prmt6. (Right) Chromatin-immunoprecipitation assays of AR occupancy at androgen-responsive elements (red bar) in C2C12 myoblasts expressing AR24Q or AR100Q treated with vehicle or DHT (10 nM, 12 h). Shown is one experiment representative of three technical replicates. Graphs show mean ± SEM; two-way ANOVA followed by Tukey HSD tests (**a**, **c**, **d**), or two-tailed Student *t*-test (**b**, **e**–**h**). Source data are provided as a Source data file.

---

previous work[20]. AR24Q and AR100Q formed a complex with endogenous LSD1 in a DHT-independent manner (Fig. 2a, Supplementary Fig. 3d).

Although LSD1 is a co-activator of normal AR, whether it also co-activates polyQ-expanded AR is unclear. To address this question, we assessed AR activity by expressing AR24Q and AR65Q driven by the cytomegalovirus promoter and measured the activity of a luciferase reporter under the control of an ARE, as done previously[34]. We applied both GOF (overexpression) and LOF (knock-down) approaches. In HEK293T cells, LSD1 overexpression significantly enhanced AR24Q and AR65Q transactivation by 1.4-fold in DHT-treated cells, and it had no effect in vehicle-treated cells, indicating that LSD1 works as a transcriptional co-activator for AR (Fig. 2b). Similar results were obtained by expressing AR12Q and AR55Q under the control of the eukaryotic promoter, elongation factor 1α (Supplementary Fig. 4a). Further, LSD1-2a, LSD1-8a, and LSD1-2a/8a transactivated normal and polyQ-expanded AR to a similar extent as LSD1 in HEK293T cells (Supplementary Fig. 4b) and MN1 cells (Fig. 2c, Supplementary Fig. 4c).

To suppress endogenous LSD1, we used CRISPR/Cas9 technology[35]. Using Cas9 with different single gRNAs in HEK293T cells, we obtained partial knock-down of *LSD1*, while concomitant use of two gRNAs produced a large in-frame deletion of *LSD1* exons encoding essential catalytic regions, thus producing an enzymatically dead LSD1 fragment (Fig. 2d, Supplementary Fig. 4d, Supplementary Table 2). Partial (50–70%) and complete knock-out of *LSD1* reduced AR24Q and AR65Q transactivation by 40–50% and 80%, respectively, indicating a dose-dependent effect of endogenous LSD1 on androgen-induced AR transactivation (Fig. 2d). Consistently, polyQ-expanded AR transactivation also was reduced upon *LSD1* knock-down, although it was significantly lower only upon complete *LSD1* knock-out. In addition, CRISPR-Cas9 knock-down of *Lsd1* significantly diminished AR24Q and AR100Q transactivation in MN1 cells (Fig. 2e). Collectively, these results show that LSD1 forms a complex with and co-activates polyQ-expanded AR.

### LSD1 requires the AR activating function-2 surface and its catalytic activity to transactivate AR

Both LSD1 and PRMT6 have an LXXLL motif (Fig. 3a)[20], which mediates the interaction of transcription co-factors with steroid receptors through the activating function-2 (AF-2) surface in the ligand-binding domain[36]. Consistently, LSD1 failed to transactivate AR bearing the E897K (where E is glutamic acid and K is lysine) mutation that disrupts co-factor recruitment through AF-2 (Fig. 3b, Supplementary Fig. 5a), similarly to PRMT6[20]. Conversely, LSD1 with a defective LXXLL motif (mutated to LXXAA, where A is alanine) also failed to transactivate AR (Fig. 3c, Supplementary Fig. 5b). We previously showed that PRMT6 catalytic activity is required for AR transactivation[20]. The catalytic-inactive LSD1 mutant K685A failed to transactivate normal and polyQ-expanded AR (Fig. 3d, Supplementary Fig. 5c). Further, treatment of mock-transfected cells with the LSD1 catalytic inhibitor

tranylcypromine (TCP) reduced polyQ-expanded AR transactivation induced by DHT (Fig. 3e); treatment of cells with the selective LSD1 inhibitor SP-2509, which inhibits the association of LSD1 with CoREST but does not affect LSD1 enzymatic activity, did not modify polyQ-expanded AR transactivation, further supporting the relevance of LSD1 catalytic activity for AR transactivation. Notably, we found significantly decreased histone 3 lysine 4 dimethylation (H3K4me2) in C2C12 myotubes expressing AR100Q, confirming that LSD1 is overactivated in SBMA myotubes (Fig. 3f). Taken together, these observations indicate that LSD1 requires its catalytic activity and the AR AF-2 surface to transactivate both normal and polyQ-expanded AR through a mechanism that involves histone modification.

### LSD1 and PRMT6 interact to cooperatively enhance AR transactivation

The observation that PRMT6 and LSD1 are both co-activators of AR and are overexpressed in SBMA skeletal muscle prompted us to test whether they synergistically contribute to the toxic GOF of polyQ-expanded AR. We first addressed whether LSD1 and PRMT6 interact with each other in HEK293T cells, after verifying that neither overexpression nor silencing of *PRMT6* modified expression of endogenous and exogenously expressed LSD1 and vice versa (Supplementary Fig. 6a). PRMT6 immunoprecipitation pulled down LSD1, indicating that PRMT6 and LSD1 interact in HEK293T cells. Further, PLA in MN1 cells expressing AR24Q and AR100Q showed that endogenous PRMT6 interacted with endogenous LSD1, and this interaction was not significantly modified by DHT (Fig. 4a).

The observation that LSD1 and PRMT6 interact with each other and with AR through the AF-2 surface led us to hypothesize that they are both required to form a functional complex with AR. We tested this hypothesis in MN1 *Lsd1* CRISPR-Cas9 knock-down cells. To silence endogenous *Prmt6*, we transduced cells with lentivirus expressing scramble short-hairpin RNA (shRNA) or two shRNAs against *Prmt6* (#1 and #2)[37]. Western blotting verified that knock-down of either *Lsd1* or *Prmt6* did not modify AR, PRMT6, and LSD1 levels (Fig. 4b). We assessed protein–protein interactions by PLA in MN1 cells. Strikingly, the interaction of normal and polyQ-expanded AR with LSD1 was significantly diminished upon *Prmt6* silencing, as was the interaction of AR with PRMT6 upon *Lsd1* silencing, even if this interaction is aberrantly enhanced by expanded polyQ (Fig. 4a). Immunoprecipitation assay confirmed AR/LSD1/PRMT6 interaction in the quadriceps muscle of WT and knock-in SBMA mice expressing AR113Q (Fig. 4c).

Next, we assessed whether LSD1 and PRMT6 cooperatively transactivate AR. We first took a GOF approach. Overexpression of LSD1 alone, PRMT6 alone, and both together enhanced transactivation of normal AR by 1.3-, 3.1-, and 4.5-fold, respectively, and that of polyQ-expanded AR by 1.5-, 3.4-, and 6.6-fold (Fig. 4d). Notably, polyQ-expanded AR transactivation by LSD1/PRMT6 was increased compared to normal AR, consistent with previous findings[20]. We previously

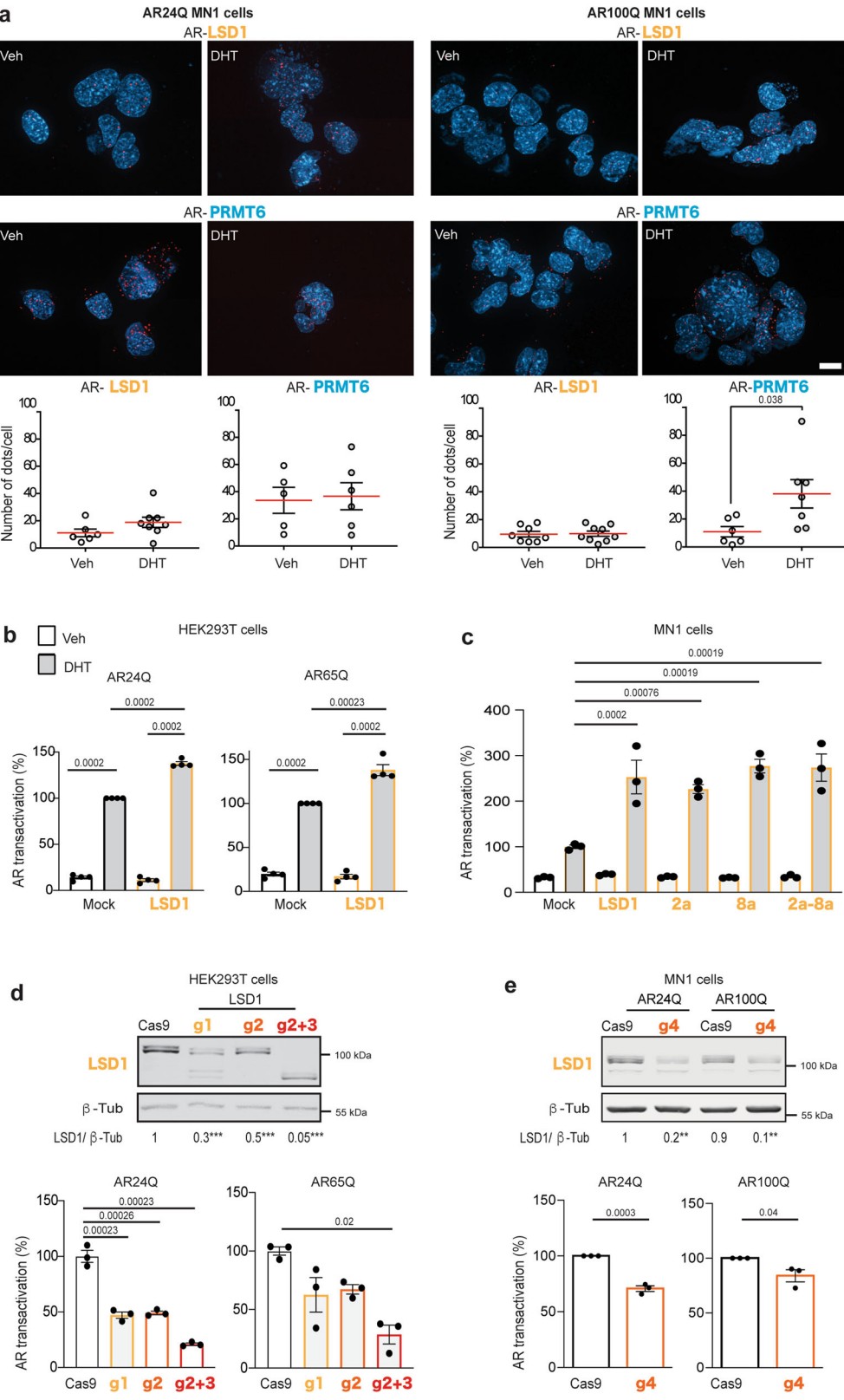

showed that AR transactivation by PRMT6 is negatively regulated by phosphorylation at AKT consensus sites, [210]RXRXXS[215] and [787]RXRXXS[792] (where R is arginine and S is serine)[20]. Interestingly, AR transactivation by LSD1 and PRMT6 was significantly enhanced by phospho-defective substitutions with alanine (S215A, S792A) compared to AR65Q with intact S215 and S792 residues, suggesting that AR transactivation by LSD1 and PRMT6 is modulated by phosphorylation at AKT consensus sites (Supplementary Fig. 6b).

We then took a pharmacological and genetic LOF approach to assess the effect of inhibition of endogenous (mock-transfected cells) and overexpressed LSD1 and PRMT6 on AR activity via transcriptional assays. TCP and the PRMT inhibitor adenosine dialdehyde (AdOx) reduced normal and polyQ-expanded AR transactivation induced by DHT (Fig. 4d), indicating that AR requires both LSD1 and PRMT6 function for full transactivation. Notably, the effect of overexpression of LSD1 on AR transactivation was reduced not only by TCP as

**Fig. 2 | LSD1 is a co-activator of polyQ-expanded AR. a** Proximity ligation assays (PLA) in MN1 cells expressing AR24Q or AR100Q treated with vehicle or DHT (10 nM, 16 h). Nuclei were detected with DAPI. Scale bar = 17 μm. Graphs show quantification of nuclei from three biological replicates (AR24Q AR/LSD1 vehicle: $n = 26$; AR24Q AR/LSD1 DHT: $n = 43$; AR24Q AR/PRMT6 vehicle: $n = 20$; AR24Q AR/PRMT6 DHT: $n = 8$; AR100Q AR/LSD1 vehicle: $n = 76$; AR100Q AR/LSD1 DHT: $n = 61$; AR100Q AR/PRMT6 vehicle: $n = 66$; AR100Q AR/PRMT6 DHT: $n = 43$). **b** Transcriptional assays in HEK293T cells expressing AR24Q or AR65Q alone (mock) or together with LSD1 treated with vehicle or DHT (10 nM, 16 h; $n = 4$ biological replicates). **c** Transcriptional assays in MN1 cells expressing AR65Q alone (mock) or together with the indicated LSD1 isoforms treated with vehicle or DHT (10 nM, 16 h; $n = 3$ biological replicates). **d** (Top) Western blots of LSD1 levels in HEK293T cells expressing Cas9 with or without specific guides to silence *LSD1* ($n = 7$ biological replicates). *** g1 $p = 0.00016$, g2 $p = 0.00017$, g3 $p = 0.00016$. (Bottom) Transcriptional assays in Cas9 and g1, g2, g2 + 3 cells expressing AR24Q or AR65Q treated with DHT (10 nM, 16 h; $n = 3$ biological replicates). **e** (Top) Western blots of LSD1 levels in MN1 cells expressing Cas9 with or without a specific guide (g4) to silence *Lsd1* ($n = 5$ biological replicates). ** g4 in AR24Q cells $p = 0.0065$, ** g4 in AR100Q cells $p = 0.0054$. (Bottom) Transcriptional assays in Cas9 and g4 cells expressing AR24Q or AR100Q and treated with DHT (10 nM, 16 h; $n = 3$ biological replicates). LSD1, PRMT6, and AR were detected with specific antibodies, and β-Tub was used as loading control. Graphs show mean ± SEM; two-tailed Student $t$-test (**a**, **e**−transcriptional assays), or two-way (**b**, **c**, **e**−Western blot), or one-way (**d**) ANOVA followed by Tukey HSD tests. Source data are provided as a Source data file.

expected but also by AdOx. Similarly, the effect of overexpression of PRMT6 was attenuated by AdOx as well as TCP. Combined treatment with TCP/AdOx further attenuated AR transactivation both in mock-transfected cells and in cells overexpressing LSD1, PRMT6, and LSD1/PRMT6. Because AdOx is a pan-PRMT inhibitor, and TCP can have additional effects on enzymes other than LSD1[38], we genetically silenced endogenous *LSD1* and *PRMT6* by CRISPR-Cas9 technology (Figs. 2d and 4e). Knock-down of *LSD1* and *PRMT6* with two independent guides alone and together attenuated normal and polyQ-expanded AR transactivation induced by DHT, demonstrating that endogenous LSD1 and PRMT6 are required for the transcriptional response of AR induced by ligand (Fig. 4e). Further, PRMT6-induced transactivation of normal and polyQ-expanded AR was significantly reduced by genetic silencing of endogenous *LSD1* (Supplementary Fig. 6c). These observations highlight an intimate relationship in normal and polyQ-expanded AR co-regulation by LSD1 and PRMT6.

### LSD1 and PRMT6 cooperatively enhance the toxic GOF of polyQ-expanded AR

We previously showed that silencing of the *Drosophila* ortholog of *PRMT6* (*Dart8*) ameliorates the neurodegenerative phenotype caused by polyQ-expanded AR (AR52Q) in a fly model of SBMA[20]. To determine whether LSD1 modifies the SBMA phenotype and synergistically cooperates with PRMT6 to enhance toxicity in vivo, we crossed SBMA flies with flies expressing RNA interference (RNAi) targeting the *Drosophila* ortholog of *LSD1*, *dLsd1*, and *Dart8*. As control, we used flies expressing AR without the polyQ tract (AR0Q). AR0Q flies fed DHT did not show any phenotype (Fig. 5a), as previously reported[11,20]. To model SBMA, we used flies expressing AR with 52Q (AR52Q). AR52Q flies developed degeneration of the posterior side of the eye, as shown before[11,20]. Silencing of *dLsd1* by ~40% did not modify polyQ-expanded AR toxicity, whereas silencing of *Dart8* by about 50% had a significant yet modest effect on polyQ-expanded AR toxicity (Fig. 5a–c), as previously reported[20]. Consistent with the synergistic effect on polyQ-expanded AR transactivation, silencing of both *dLsd1* and *Dart8* together strongly suppressed the DHT-induced degenerative phenotype caused by polyQ-expanded AR. This evidence supports the idea that LSD1 and PRMT6 synergistically cooperate to enhance polyQ-expanded AR toxic GOF in vivo.

### Artificial miRNA strategy to silence mouse *Lsd1* and *Prmt6*

The results described above prompted us to develop a therapeutic strategy to target AR co-regulators. Gene silencing can be pursued using of artificial microRNAs (miRNAs) or shRNAs[39–41]. miRNAs can be expressed at lower levels compared to shRNAs using RNApol II promoters that allow high expression in target cells yet ensure efficient processing and no neuronal damage. Moreover, miRNAs target a specific protein within the same family and are thus highly specific with respect to small molecule inhibitors. We designed 83 and 51 artificial miRNAs (amiRs) to silence mouse *Lsd1* and *Prmt6*, respectively, with the BLOCK-iT RNAi Designer. Using the best alignments targeting all known mouse *Lsd1* isoforms and *Prmt6*, we selected top-ranking amiRs

for in vitro validation (Fig. 6a, Supplementary Table 3). In MN1 cells, amiR-*Lsd1*#1 silenced *Lsd1* expression by 50%, whereas amiR-*Lsd1*#2 and #3 silenced expression by 70%. Because *Lsd1* knock-out is embryonic lethal, whereas its haploinsufficiency is not associated with major consequences in mouse[42], we selected amiR-*Lsd1*#1 for further analysis. Three of seven *Prmt6* amiRs (#2, #6, and #7) significantly silenced *Prmt6* to a similar extent (~50%), and we pursued amiR-*Prmt6*#6 for further analysis. MN1 cells expressing AR100Q, which showed reduced viability compared to MN1 cells expressing AR24Q in the presence of DHT, had significantly increased cell viability upon simultaneous silencing of both *Lsd1* and *Prmt6* (Fig. 6b), consistent with results from SBMA flies.

To move in vivo, we used an adeno-associated virus subtype 9 (AAV9) that expresses green fluorescent protein (GFP) (Fig. 6c). A similar strategy has recently been successfully applied in transgenic SBMA mice expressing AR100Q[32]. One of the most important features of our amiR-based strategy for target gene silencing and AAV9 delivery is the possibility to clone more than one amiR in the same vector, which allowed us to use the same amount of virus to target both genes. To set up the conditions of viral infection, we performed a single intraperitoneal injection of amiR-*Lsd1*/*Prmt6* at concentrations of $2^{10}$ and $8^{10}$ virion particles at 3 weeks of age. By analyzing the biodistribution of AAV9 expressing control amiR, we detected viral particles in several tissues including liver, heart, quadriceps, and spinal cord (Fig. 6d). Western blotting detected GFP expression in several tissues including skeletal muscle (Fig. 6e). Transduction of amiR-*Lsd1*/*Prmt6* in AR100Q mice significantly reduced *Lsd1* and *Prmt6* transcript levels, with no effect on mouse and human *AR* transcript levels, in the skeletal muscle (Fig. 6f). Western blotting confirmed significantly reduced expression of LSD1 and PRMT6 proteins in skeletal muscle, but not spinal cord, liver, white adipose tissue, and lungs, although LSD1 was significantly reduced in the heart (Fig. 6g and Supplementary Fig. 7). As LSD1 and PRMT6 are epigenetic writers that modify gene expression and AR is a transcription factor active in several tissues, we measured transcript levels of AR, LSD1, and PRMT6 target genes in tissues other than skeletal muscle. amiR-*Lsd1*/*Prmt6* did not significantly modify expression of selected target genes in most tissues, thus excluding a general effect on gene transcription (Supplementary Fig. 8)[20,32,43,44]. These observations show that AAV9-mediated delivery of amiR-*Lsd1*/*Prmt6* is an effective strategy to silence AR co-regulators in vivo without affecting *AR* expression.

### amiR-*Lsd1*/*Prmt6* reduces gene expression dysregulation in skeletal muscle of SBMA mice

To translate our findings into therapy for SBMA patients, we tested the efficacy of our amiRs in AR100Q mice. We first explored whether knock-down of LSD1 and PRMT6 attenuates disease-related transcriptional dysregulation. RNA-seq transcriptomic analysis in quadriceps muscle of 13-week-old control and amiR-*Lsd1*/*Prmt6*-treated AR100Q and WT mice identified 6583 differentially expressed genes (DEGs) (4158 upregulated and 2425

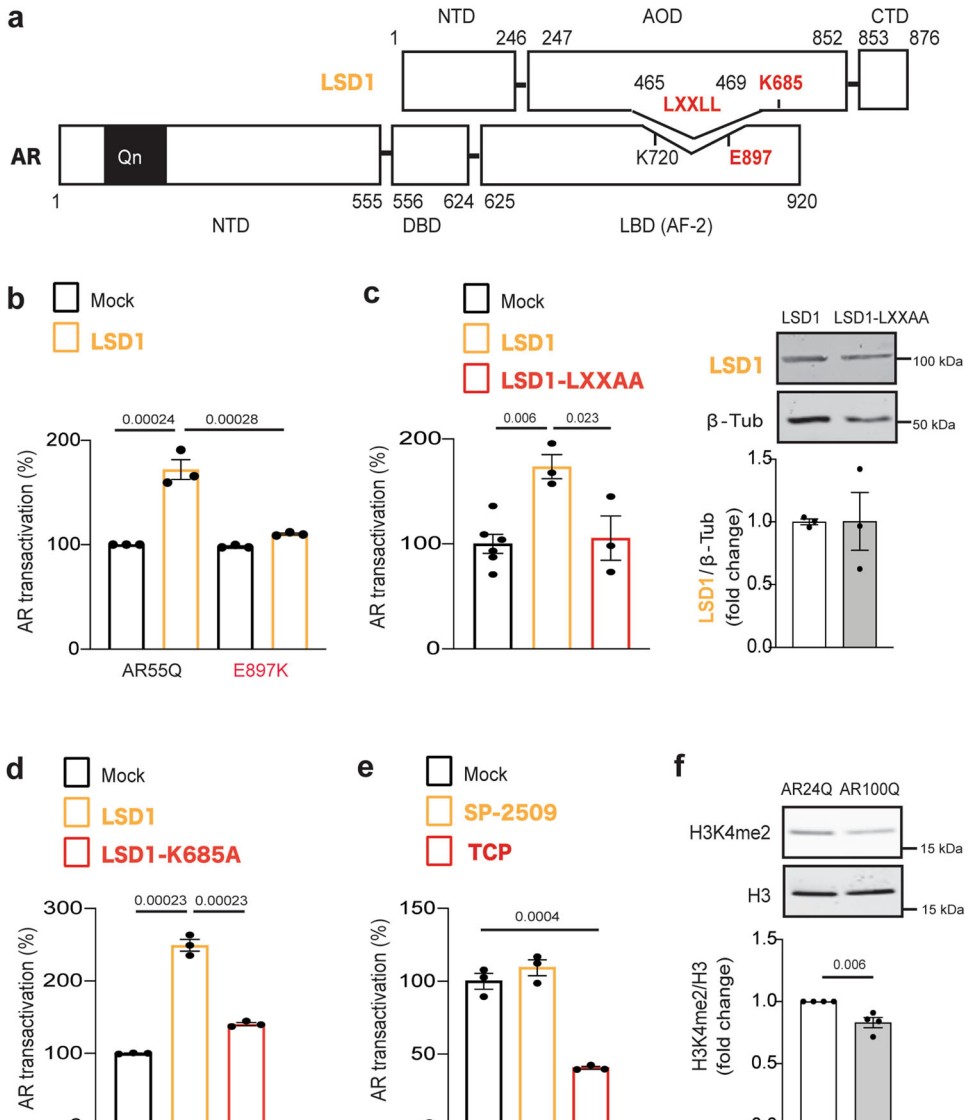

**Fig. 3 | LSD1 requires the AR AF-2 surface and its catalytic activity to transactivate AR. a** Scheme of AR and LSD1 modular domains and specific motifs. Numbers refer to AR NM_000044 and LSD1 NM_001009999. NTD, amino-terminal domain; DBD, DNA-binding domain; LBD, ligand-binding domain; AOD, amine oxidase domain; CTD, carboxy-terminal domain. **b** Transcriptional assay in HEK293T cells expressing AR55Q or the AF-2 mutant AR55Q-E897K, alone (mock) or with LSD1. Cells were treated with DHT (10 nM, 16 h; *n* = 3 biological replicates). **c** (Left) Transcriptional assay in HEK293T cells expressing AR65Q alone (mock) or with either LSD1 or LSD1-LXXAA. Cells were treated with DHT (10 nM, 16 h; *n* = 6 biological replicates mock, *n* = 3 biological replicates LSD1 and LSD1-LXXAA). (Right) Western blot of LSD1 and LSD1-LXXAA expression in HEK293T cells. Shown

is one representative image of three biological replicates. **d** Transcriptional assay in HEK293T cells expressing AR65Q alone (mock) or with either LSD1 or the catalytic inactive mutant LSD1-K685A. Cells were treated with DHT (10 nM, 16 h; *n* = 3 biological replicates). **e** Transcriptional assay in HEK293T cells expressing AR65Q treated with DHT only or together with SP-2509 (100 nM) or TCP (10 µM) for 16 h (*n* = 3 biological replicates). **f** Western blot of H3K4me2 in C2C12 cells differentiated to myotubes for 10 days in the presence of DHT (10 nM). H3K4me2 was detected with a specific antibody that recognize the H3 modified K residue. H3 antibody was used as loading control (*n* = 4 biological replicates). Graphs show mean ± SEM; two-way (**b**), one-way (**c**, **d**, **e**) ANOVA followed by Tukey HSD tests, or two-tailed Student *t*-test (**f**). Source data are provided as a Source data file.

downregulated genes, GSE193539) in SBMA muscles compared to WT muscles (absolute log2 fold change >2, corrected *p* < 0.01) (Fig. 7a, Supplementary Data 1), consistent with recent observations of a high number of many altered genes in the tibialis anterior muscle at 11 weeks in the same SBMA mouse model[32]. When we analyzed the effect of amiR treatment in AR100Q mice, we found 1129 DEGs (488 upregulated and 641 downregulated genes). Of note, 285 genes were completely rescued, showing transcriptional levels comparable to WT mice, while 389 genes demonstrated partial rescue (Fig. 7b). No DEGs were observed when comparing amiR-treated WT to control WT muscles.

We performed functional enrichment analysis on the partially and totally rescued genes and obtained statistical significance for several

terms including "skeletal muscle contraction," "muscle structure development," "myofibril assembly," "sarcoplasmic reticulum calcium ion transport," "collagen chain trimerization," "myofibril assembly and metabolism," "fructose glycogen metabolism," "protein nitrosylation," "generation of precursor metabolites and energy," and "purine nucleoside monophosphate metabolic process" (Fig. 7c, d). Interestingly, most (~80%) of GO biological processes previously identified as enriched in muscles of female mice expressing polyQ-expanded AR and treated with androgens[45] had an overlap coefficient of ≥0.5 with the GO biological processes enriched by our rescued genes (65% with overlap coefficient of 1). This suggests that genes in these categories are transcriptionally dysregulated by mutant AR and partially or totally normalized upon *Lsd1* and *Prmt6* silencing.

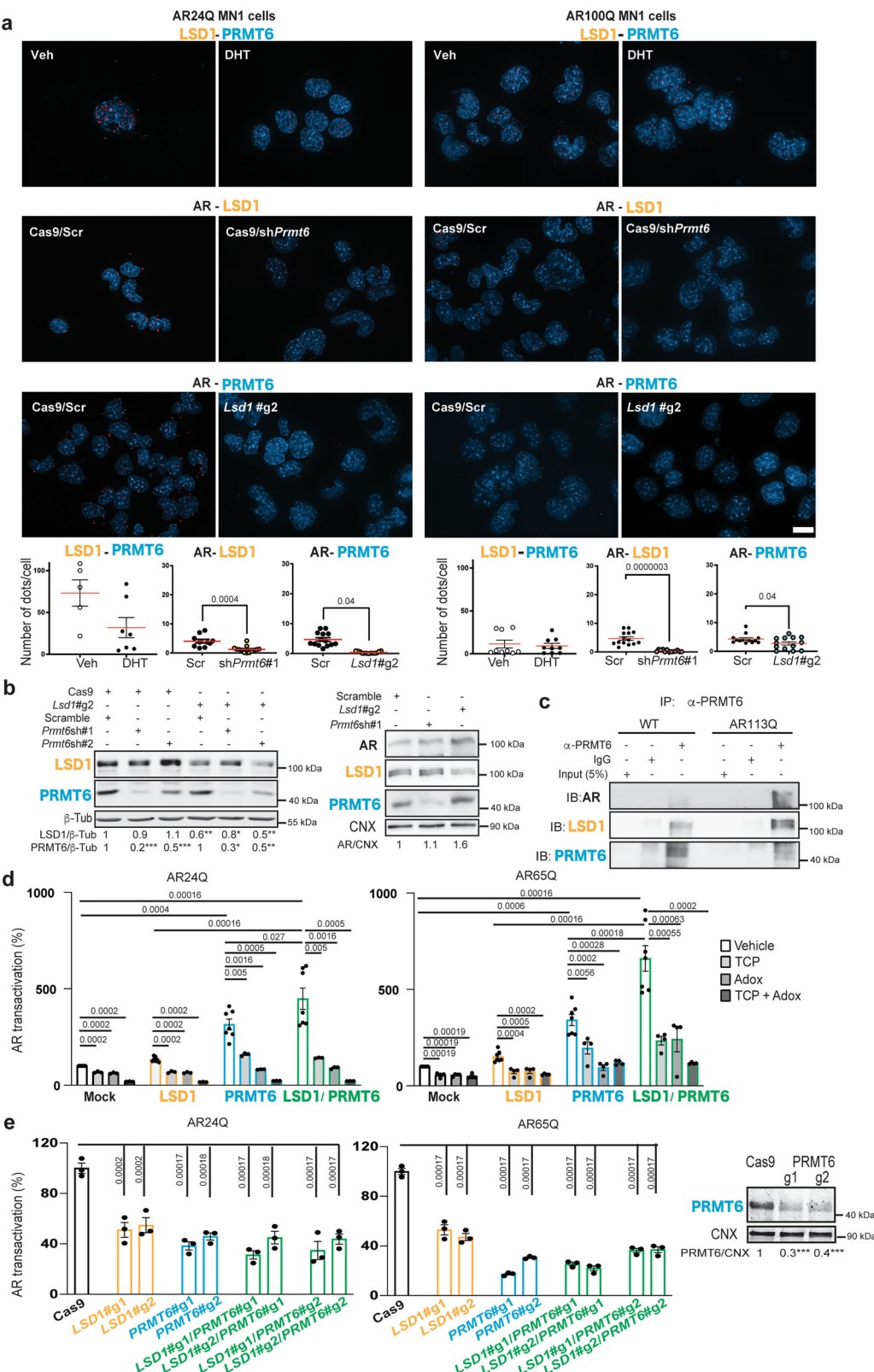

We then compared our data with published transcriptomic analyses on SBMA muscles expressing either AR113Q (knock-in model), AR97Q (another transgenic model), or WT AR only in the muscle (HSA-AR)[46]. We checked how many of the genes dysregulated in the three models were rescued by our treatment. *Lsd1* and *Prmt6* silencing rescued 52 of 153 (34%) genes dysregulated in the HSA-AR model. Similarly, 58 of 204 (28%) and 31 of 159

(20%) genes dysregulated in the AR97Q and AR113 models were rescued, respectively. We also considered whether treatment modified expression of genes dysregulated in a LOF model of AR[47]. Only 17% of these genes also were rescued by amiR treatment, showing a limited reduction of AR physiological functions. Overall, the data demonstrate an effect on polyQ-expanded AR toxic GOF.

**Fig. 4 | LSD1 and PRMT6 synergistically transactivate normal and polyQ-expanded AR. a** Proximity ligation assays analysis in MN1 cells expressing AR24Q or AR100Q and vectors for *Lsd1* and *Prmt6* silencing. Cells were treated with DHT (10 nM, 16 h). Scale bar = 17 μm. Graphs show quantification of nuclei from three independent experiments (AR24Q cells: P6/LSD1 vehicle $n = 12$, DHT $n = 22$; AR/LSD1 Cas9 scrambled $n = 169$, Cas9 shPrmt6 $n = 198$; AR/PRMT6 Cas9 scrambled $n = 184$, Cas9 g2 $n = 237$. AR100Q cells: P6/LSD1 vehicle $n = 79$, DHT $n = 116$; AR/LSD1 Cas9 scrambled $n = 302$, Cas9 shPrmt6 $n = 190$; AR/PRMT6 Cas9 scrambled $n = 202$, Cas9 g2 $n = 164$. **b** Western blots of MN1 cells transduced with lentiviral vectors for *Lsd1* and *Prmt6* silencing ($n = 3$ biological replicates). LSD1 g2 **$p = 0.0036$, g2/sh#1 *$p = 0.048$, g2/sh#2 **$p = 0.0071$. PRMT6 sh#1 ***$p = 0.00001$, sh#2 ***$p = 0.0000004$, g2/sh#1 *$p = 0.011$, g2/sh#2 *$p = 0.003$. **c** Immunoprecipitation of PRMT6 and immunoblotting of the indicated proteins in the skeletal muscle

(quadriceps) of 24-week-old WT and knock-in AR113Q mice ($n = 3$ mice/genotype). **d** Transcriptional assays in HEK293T cells expressing AR24Q or AR65Q alone (mock) or with LSD1 and PRMT6 treated for 16 h with vehicle, DHT (10 nM), TCP (10 μM), or Adox (10 μM) (AR24Q: $n = 7$ vehicle, $n = 3$ TCP, Adox, TCP + Adox. AR65Q: $n = 7$ vehicle, $n = 4$ TCP, Adox, TCP + Adox). **e** (Left) Transcriptional assay in HEK293T cells expressing AR24Q or AR65Q with or without CRISPR guides to silence *Lsd1* and *Prmt6* ($n = 3$ biological replicates). (Right) Western blot of PRMT6 levels in HEK293T expressing Cas9 alone or together with guides to silence *PRMT6* ($n = 3$ biological replicates). Cas9/g1 ***$p = 0.00048$, Cas9/g2 ***$p = 0.00056$. AR, LSD1, and PRMT6 were detected with specific antibodies, and β-Tub and CNX were used as loading controls. Graphs show mean ± SEM; two-tailed Student *t*-test (**a**, **b**) or one-way ANOVA followed by Tukey HSD tests (**d**, **e**). Source data are provided as a Source data file.

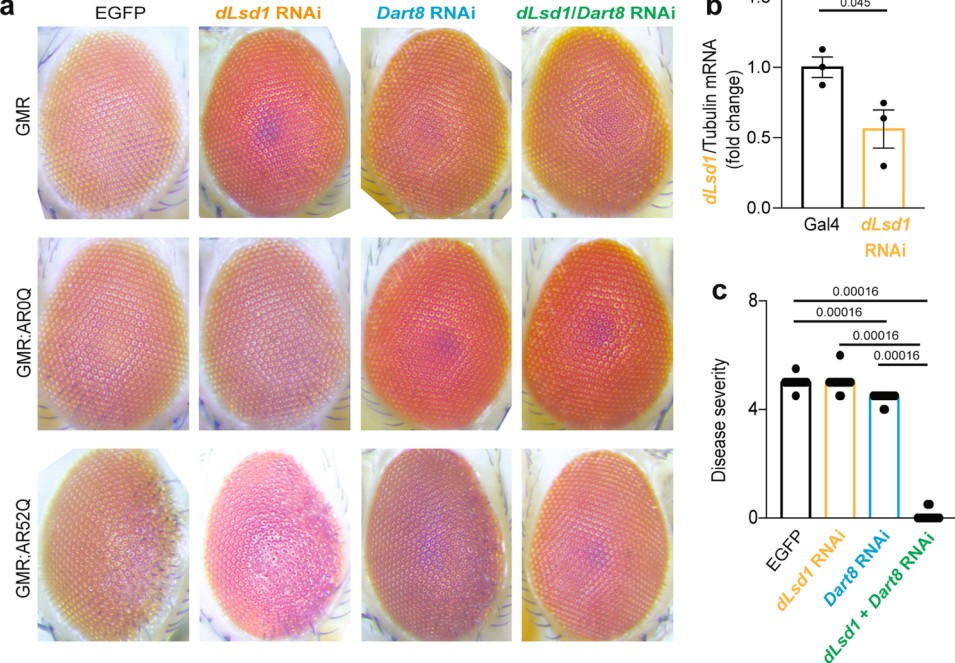

**Fig. 5 | Silencing of LSD1 and PRMT6 together suppresses polyQ-expanded AR neurotoxicity. a** Analysis of the eye phenotype in flies expressing EGFP, AR0Q, or AR52Q with or without RNAi to silence *dLsd1* and the PRMT6 fly ortholog *Dart8*. Shown are representative images from 10–15 flies/genotype. **b** RT-PCR analysis of *dLsd1* mRNA transcript levels normalized to *tubulin* ($n = 3$ flies/genotype). **c** Disease

severity in AR52Q flies with or without *dLsd1* and *Dart8* silencing ($n = 15$ AR52Q; $n = 16$ AR52Q *dLsd1* RNAi and AR52Q *Dart8* RNAi; $n = 12$ AR52Q *dLsd1* RNAi/*Dart8* RNAi). Graphs show mean ± SEM; two-tailed Student *t*-test (**b**) or one-way ANOVA followed by Tukey HSD test (**c**). Source data are provided as a Source data file.

## amiR-*Lsd1*/*Prmt6* treatment ameliorates the disease phenotype of SBMA mice

We performed a preclinical study to assess the effect of *Lsd1*/*Prmt6* silencing on the mouse disease phenotype. We randomized a cohort of male transgenic AR100Q mice and WT siblings and assigned them to either control amiR or amiR-*Lsd1*/*Prmt6* treatment groups. We first verified that our gene therapy strategy did not modify body weight and motor function of WT mice to exclude undesired effects from genetic manipulation of target genes (Fig. 8a). This was particularly important for *Lsd1*, which is embryonic lethal when deleted in mice[42]. We then analyzed the effect of treatment on the SBMA phenotype. AR100Q mice show reduced body weight and motor dysfunction starting around 8 weeks of age[28]. Treatment with amiR-*Lsd1*/*Prmt6* slightly yet significantly increased their body weight at 9–11 weeks of age and significantly improved muscle strength by hanging wire task and motor coordination by rotarod tasks (Fig. 8a).

SBMA muscle is characterized by a glycolytic-to-oxidative fiber-type switch[16]. Consistent with the role of LSD1 on muscle metabolism[48],

amiR-*Lsd1*/*Prmt6* reduced the number of oxidative fibers from 49% in vehicle-treated mice to 37% in amiR-*Lsd1*/*Prmt6*-treated mice (Fig. 8b). A key aspect of skeletal muscle pathology in SBMA is functional denervation associated with upregulation of genes, such as muscle associated receptor tyrosine kinase (*Musk*), myogenin (*MyoG*), and neural cell adhesion molecule (*NCAM*), which are induced upon dysfunctional communication between the motor neuron and innervated myofiber[10,16,28]. *Musk*, *MyoG*, and *NCAM* were upregulated in the muscle of AR100Q mice and were significantly reduced by amiR-*Lsd1*/*Prmt6* treatment (Fig. 8c). This is noteworthy because these genes are regulated by LSD1[48–51]. AR100Q forms 2% SDS-resistant aggregates in muscle, which can be detected as high molecular weight species that accumulate in the stacking portion of a polyacrylamide gel[28]. Consistent with our previous findings that PRMT6 enhances polyQ-expanded AR aggregation[20], western blotting showed that silencing *Lsd1* and *Prmt6* reduced accumulation of high molecular weight species and monomeric AR in muscle (Fig. 8d). Together, these results show that silencing *Lsd1* and *Prmt6* attenuates the severe disease phenotype in a murine model of SBMA.

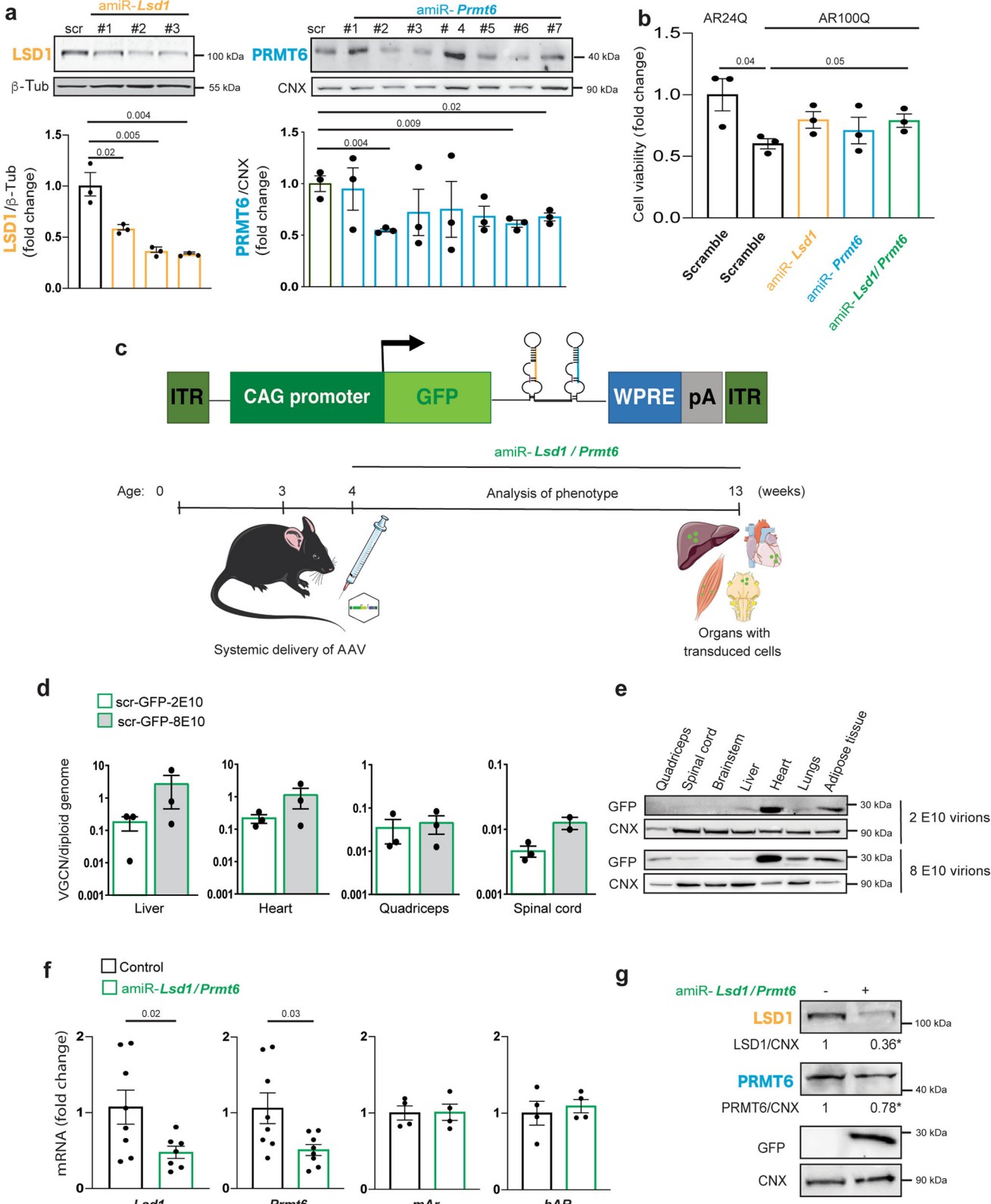

**amiRs targeting human *LSD1*/*PRMT6* reduce proliferation of an androgen-sensitive prostate cancer cell line**

To validate our strategy in human cells, we designed three amiRs to silence human *LSD1* and *PRMT6* (Supplementary Table 3). We verified target silencing in HEK293T cells and in the androgen-sensitive prostate cancer cell line, LNCaP, because LSD1 and PRMT6 are overexpressed in prostate cancer and correlate with

cancer aggressiveness (Fig. 9a)[22,23,27]. In HEK293T cells, silencing of both *LSD1* and *PRMT6* modified the expression of two genes regulated by AR and its co-regulators: *ATP2A2*, coding for SERCA2; and *CDKN1A*, coding for p21[CIP1] (Fig. 9b)[20,44,52]. In LNCaP cells, we observed significantly reduced cell proliferation with targeting of either *LSD1* or *PRMT6*, showing that these amiRs exert a biological effect in prostate cancer cells (Fig. 9c). These

**Fig. 6 | miRNA strategy to silence *Lsd1* and *Prmt6* in vivo. a** Western blots of LSD1 and PRMT6 in MN1 cells transfected with vectors expressing scramble amiR or amiR to silence *Lsd1* and *Prmt6*. Shown is one experiment representative of three biological replicates. **b** Cell viability assay in MN1 cells expressing either AR24Q or AR100Q transfected as indicated (amiR-*Lsd1*#1, amiR-*Prmt6*#6) and treated with DHT (10 nM, 48 h; *n* = 3 biological replicates). **c** Schematic of AAV9 vector expressing GFP and amiR-*Lsd1*#1/*Prmt6*#6 (amiR-*Lsd1*/*Prmt6*). ITR, inverted terminal repeat; WPRE, woodchuck hepatitis virus post-transcriptional regulatory element; pA, poly-adenylation site. Figure was in part generated using pictures from Servier Medical Art licensed under a Creative Commons Attribution 3.0 Unported License (https://creativecommons.org/licenses/by/3.0/) and in part created with Biorender.com. **d** Biodistribution of viral particles in different tissues of 4-week-old WT mice (Scr-GFP-2E10 virions: *n* = 3 male mice; Scr-GFP-8E10 virions: *n* = 3 male mice for liver, heart and quadriceps; *n* = 2 male mice for spinal cord). **e** Western blots of

GFP expression in the indicated tissues of WT mice transduced by AAV9-amiR. Shown is one experiment representative of three biological replicates in three mice. **f** RT-PCR analysis of transcript levels of *Lsd1*, *Prmt6*, mouse *Ar* (*mAr*), and human *AR* (*hAR*) in the quadriceps muscle of 13-week-old AR100Q mice with or without amiR-*Lsd1*/*Prmt6* (*Lsd1 n* = 8 AR100Q and *n* = 7 AR100Q amiR-*Lsd1*/*Prmt6*; *Prmt6* expression: *n* = 8 AR100Q mice and *n* = 8 AR100Q amiR-*Lsd1*/*Prmt6*; *mAr* and *hAR*: *n* = 4 mice/treatment). **g** Western blots of LSD1 and PRMT6 in the skeletal muscle of 13-week-old AR100Q mice treated with or without amiR-*Lsd1*/*Prmt6*. Quantification is shown at the bottom. *LSD1 *p* = 0.03, *PRMT6 *p* = 0.023 (LSD1 expression: *n* = 6 mice/treatment; PRMT6 expression: *n* = 3 mice/treatment). Shown is one representative experiment. GFP, LSD1, and PRMT6 were detected with specific antibodies, and β-Tub and CNX were used as loading controls. Graphs show mean ± SEM; two-tailed Student *t*-test. Source data are provided as a Source data file.

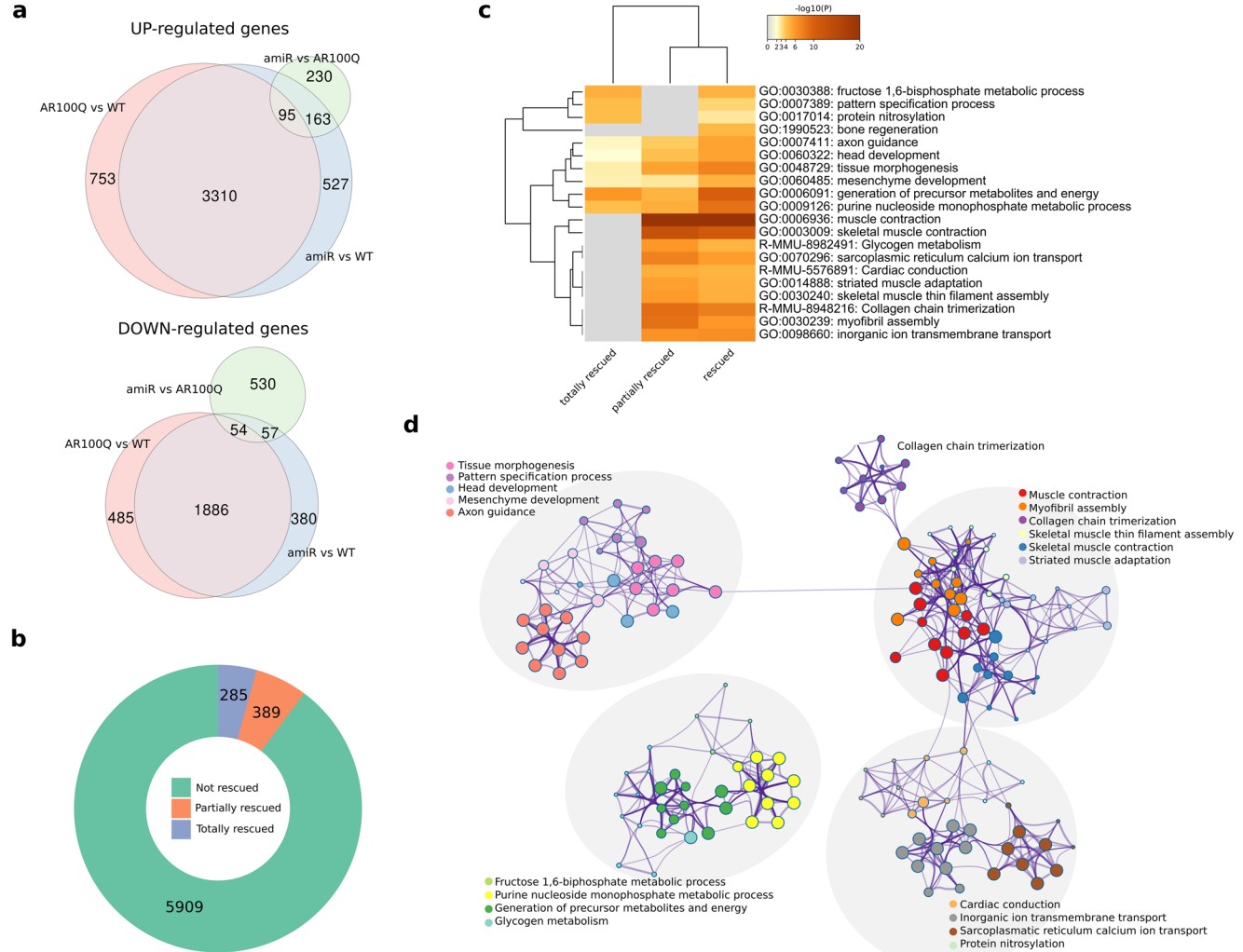

**Fig. 7 | Silencing of *Lsd1* and *Prmt6* modifies gene expression in SBMA muscle. a** Venn diagrams showing intersection of differentially expressed genes (top, upregulated; bottom, downregulated; absolute fold-change >4 and adjusted *p* < 0.01) obtained by comparing control (AR100Q) versus amiR-treated SBMA mice; AR100Q versus WT mice; or amiR-treated versus WT mice. **b** Ring plot showing the number of differentially expressed genes in AR100Q versus WT mice from differential expression analysis and relative proportions of totally and

partially rescued genes. **c** Heatmap of top 20 clusters obtained from functional enrichment analysis of rescued genes. Color is proportional to enrichment *p*-values. Each cluster name is based on the most statistically significant term within the cluster. **d** Similarity network of enriched terms. Each node represents an enriched term and is colored by cluster identifier. Node size is proportional to number of rescued genes in the term. Edge width is proportional with similarity score computed between pairs of nodes.

observations validate amiR-mediated silencing of two fundamental AR co-regulators in human cells, suggesting a potential translation of this strategy to SBMA patients and possibly other AR-associated GOF diseases, such as prostate cancer and cancer types with a sex bias.

## Discussion

Here we show that polyQ-expanded AR aberrantly enhances the expression of its own co-regulators in SBMA skeletal muscle, and targeting these overexpressed co-regulators is a strategy to suppress or attenuate AR toxic GOF (Fig. 10). SBMA is an X-linked disease,

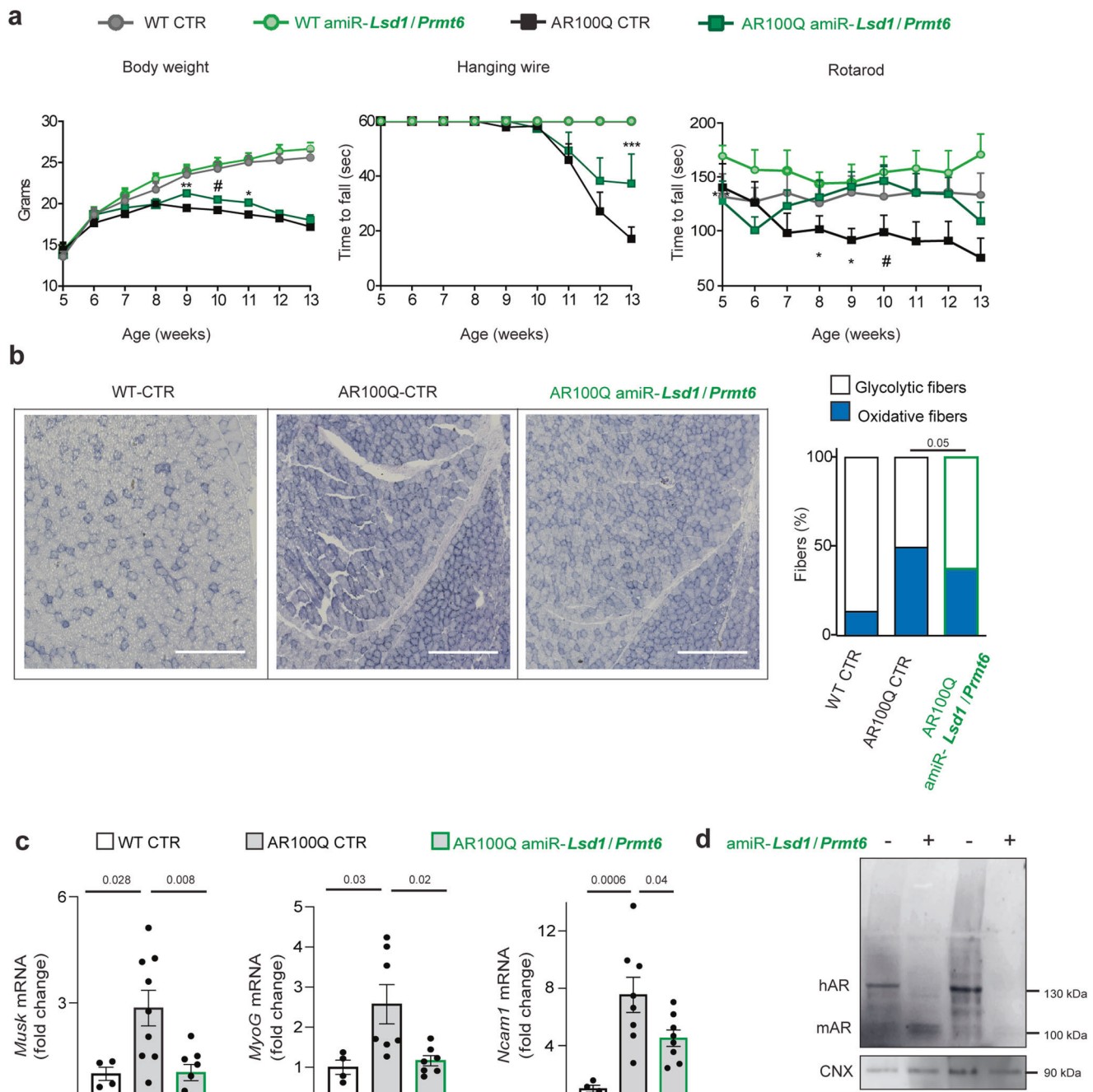

**Fig. 8 | Silencing of *Lsd1* and *Prmt6* ameliorates the disease phenotype of SBMA mice. a** Body weight, hanging wire, and rotarod analysis of male WT and AR100Q mice treated with either vehicle or amiR-Lsd1/Prmt6 (*n* = 8 WT CTR, *n* = 10 WT amiR-*Lsd1*/*Prmt6*, *n* = 9 AR100Q CTR, *n* = 8 AR100Q amiR-*Lsd1*/*Prmt6* mice. **b** NADH staining of 8-week-old mice treated as indicated (*n* = 1 WT CTR, *n* = 4 AR100Q CTR and *n* = 4 AR100Q amiR-*Lsd1*/*Prmt6*; *n* = 11,000 counted fibers/group. Scale bar = 4 μm. **c** RT-PCR analysis of denervation markers in 13-week-old WT CTR, AR100Q CTR, and AR100Q amiR-*Lsd1*/*Prmt6* mice (*n* = 4 WT CTR, *n* = 9 AR100Q CTR, *n* = 8 AR100Q amiR-*Lsd1*/*Prmt6*. **d** Western blots of AR in the skeletal muscle of AR100Q

CTR and AR100Q amiR-*Lsd1*/*Prmt6* mice (*n* = 3 mice/group). HMW, high molecular weight species. AR was detected with specific antibody and CNX was used as loading control. Graphs show mean ± SEM; two-way ANOVA followed by Fisher's least significance difference test (**a**, Body weight: *p* = 0.008 week 9, *p* = 0.081 week 10, *p* = 0.045 week 11. Hanging wire: *p* = 0.058 week 12, *p* = 0.001 week 13. Rotarod: *p* = 0.036 week 9, *p* = 0.045 week 10, p = 0.057 week 11). *\**p* < 0.05; *\*\**p* < 0.01; *\*\*\**p* < 0.001, # marginally significant; two-tailed Student t-test (**b**); one-way ANOVA followed by Tukey HSD tests (**c**). Source data are provided as a Source data file.

develops in males, and is characterized by symptoms associated with both toxic GOF and LOF mechanisms, which poses a limit to clinical approaches to silence the disease protein. This aspect is particularly relevant for chronic, slow-progressive diseases such as SBMA that require long-term treatment regimes for three reasons: (i) any therapy is more likely to work if started at puberty, concomitant with or before symptoms appear; (ii) any therapy

suppressing mutant *AR* will enhance AR LOF, an aspect that cannot be neglected in an X-linked disease affecting males; and (iii) SBMA patients show signs of androgen insensitivity syndrome, so a therapy that will be administered for the entire life of the patient must take this into account. Enhancing AR LOF is likely to exacerbate sexual dysfunction, metabolic syndrome and diabetes, depression, and muscle atrophy.

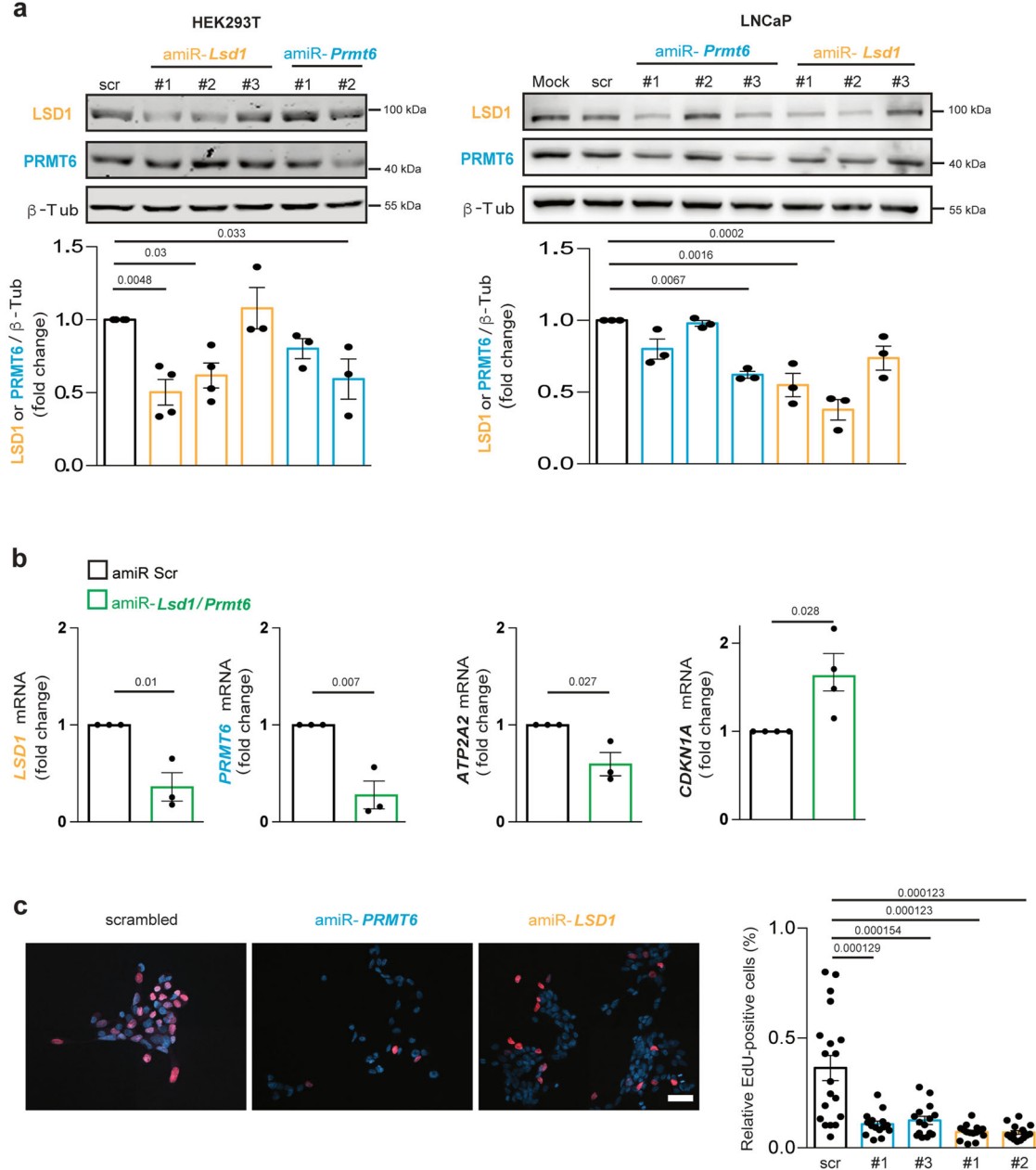

**Fig. 9 | Silencing of human *LSD1* and *PRMT6* modifies gene expression and prostate cancer cell proliferation. a** Western blots of LSD1 and PRMT6 in HEK293T and LNCaP cells transfected with vectors expressing scramble amiR or amiR-*LSD1* and amiR-*PRMT6*. Shown is one representative experiment. HEK293T cells: *n* = 4 biological replicates for scramble, amiR-*LSD1*#1 and amiR-*LSD1*#2; *n* = 3 biological replicates for amiR-*PRMT6*#1 and amiR-*PRMT6*#2. LNCaP cells: *n* = 3 biological replicates for all the conditions. Quantification of the levels of LSD1 (yellow) and PRMT6 (blue) is shown at the bottom. **b** RT-PCR analysis of the indicated genes in HEK293T transfected with vectors expressing scramble amiR or amiR-*LSD1*#1 and amiR-*PRMT6*#3 (*n* = 3 biological replicates). **c** EdU cell proliferation assay performed in LNCaP cells transduced with lentiviral vectors expressing scramble amiR or amiR-*LSD1* and amiR-*PRMT6* (*n* = 19 fields scr, *n* = 15 fields for amiRs from three independent experiments). Nuclei were detected with Hoechst® 33342. Scale bar = 53 μm. Graphs show mean ± SEM; one-way ANOVA followed by Tukey HSD tests (**a**, **c**), and two-tailed Student *t*-test (**b**). Source data are provided as a Source data file.

Co-factors are critical for the proper activity of steroid receptors. Accounting for the emerging relevant role of AR-co-regulators in disease conditions spanning from cancer to neurodegeneration, we cannot rule out the possibility that diminished expression of two key AR co-regulators at least in part enhances mutant AR LOF, even if these co-regulators are overexpressed in skeletal muscle. It is possible that in other tissues, interaction of AR with these co-regulators has a different role with different consequences than in degenerating cells. For instance, LSD1 regulates gene expression in neurons in response to neuronal activity, which suggests pathogenetic mechanisms specific to excitable cells[53]. Here we provide proof-of-principle that selectively targeting AR co-regulators that are aberrantly expressed in tissues that primarily degenerate in SBMA is a valuable therapeutic strategy.

In the unliganded state, AR mainly but not exclusively localizes to the cytosol and associates with heat shock proteins. Upon androgen binding, AR dissociates from heat shock proteins and translocates to the nucleus, where it binds to AREs. Interaction of AR with its co-regulators is necessary to assemble a functional transcription complex. At least 50 of the almost 300 known AR co-regulators are over-expressed in prostate cancer cells[54–56], which correlates with a negative outcome and prognosis[57]. Dysregulated expression of AR co-regulators contributes to onset and progression of prostate cancer and other

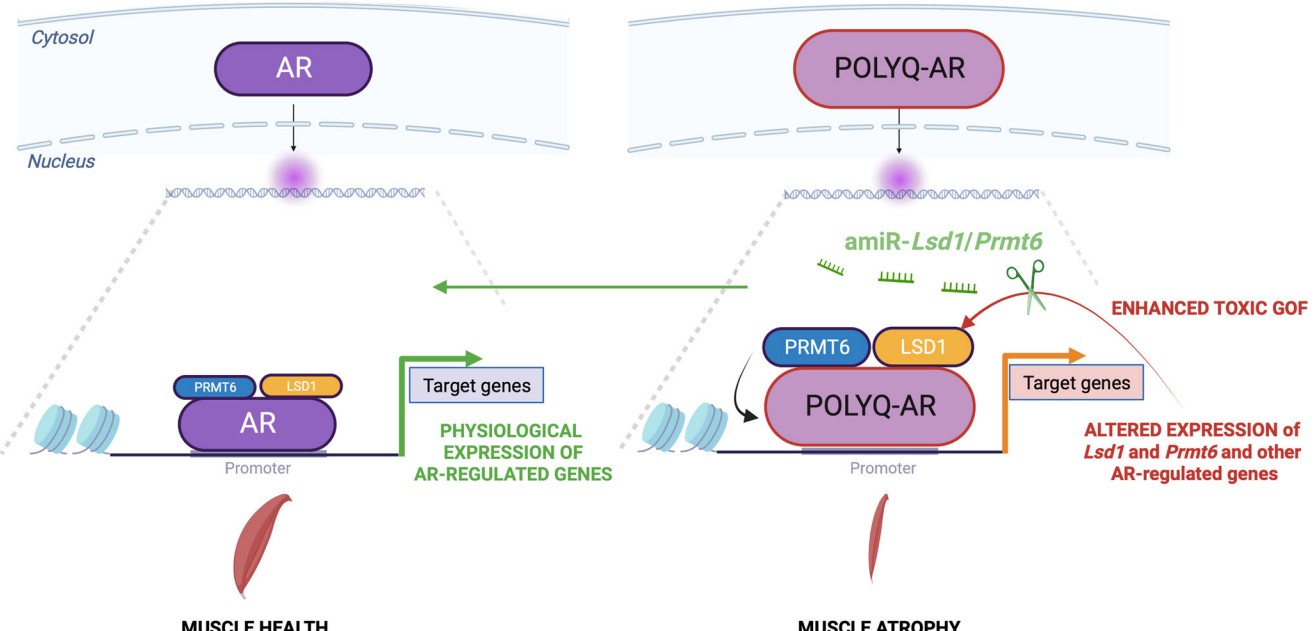

**Fig. 10 | Working model of polyQ-expanded AR and co-regulators in SBMA muscle.** In physiological conditions, AR controls expression of target genes. PolyQ expansions result in aberrant transcription of AR co-regulators, such as LSD1 and PRMT6, which in turn increase AR transactivation, thus enhancing toxic gain-of-function. Intervention to block this feedforward mechanism ameliorates disease outcome in animal models of SBMA. Created with BioRender.

types of hormone-dependent cancers, such as bladder, liver, and kidney cancers. In prostate cancer, overexpression of ~30% of AR co-regulators results from aberrant AR signaling[58,59]. In prostate cancer cells, ligand-activated AR directly promotes transcription of its co-activator Cryptochrome Circadian Regulator 1 (CRY1), which in turn regulates the expression of genes involved in DNA repair[59]. Our results highlight a similar pattern of dysregulation for two key AR co-activators in an androgen-dependent neurodegenerative disease. LSD1 and PRMT6 were aberrantly overexpressed in the skeletal muscle of male SBMA mice, while expression was much lower in female mice, and overexpression was attenuated by surgical castration. These AR co-regulators were not overexpressed in iPSC patient-derived motor neurons, the primary neuronal cells that degenerate in SBMA, as well as in liver, although SBMA patients develop nonalcoholic fatty liver[60], suggesting that the overexpression is specific to the muscle. Similar to CRY1 in prostate cancer cells, polyQ-expanded AR bound AREs present in the promoters of *Lsd1* and *Prmt6* in SBMA myotubes. Notably, binding was specific for mutant AR, occurred in the absence of ligand and was enhanced by androgens. Our results suggest a feedforward mechanism in the pathological condition, whereby polyQ-expanded AR increases expression of its own positive co-regulators, which in turn enhance AR function, leading to dysregulated expression of genes involved in muscle function.

Similar to other members of the steroid hormone receptor family, AR contains three domains: an amino-terminal transactivation domain, DNA-binding domain, and carboxy-terminal ligand-binding domain[61]. Mechanistically, both PRMT6 and LSD1 bind the AF-2 surface of AR located in the ligand-binding domain through their steroid hormone binding motif, LXXLL. Moreover, PRMT6 and LSD1 are part of the same complex and are both required to form a functional complex with AR to fully respond to androgens. Our results showed that LSD1 and PRMT6 synergistically transactivate AR. Transcription co-regulators do not directly bind DNA but rather often possess enzymatic activity and exert their function by modifying histone proteins, resulting in changes in chromatin structure, transcription factor accessibility to enhancers and promoters, transcription factor and co-factor recruitment at the poised genes, and interaction with the preinitiation complex. For example, LSD1 catalyzes the demethylation of H3K4me1/me2[62]. We found reduced H3K4me2 in C2C12 myotubes expressing polyQ-expanded AR, indicating that LSD1 is not only overexpressed, but also overactivated in SBMA myotubes.

Transcription co-factors also post-translationally target non-histone proteins involved in gene transcription. PRMT6 methylates and transactivates estrogen receptor alpha, CREB-regulated transcriptional co-activator 2, DNA topoisomerase 3B, p16[INK4a], p21[CIP1], DNA polymerase beta, and high mobility group A1a[21,63–68]. LSD1 also targets non-histone proteins, including Forkhead Box A1, DNA methyltransferase 1, p53, hypoxia-inducible factor alpha, estrogen-related receptor alpha, and E2F[42,69–74]. Androgen binding induces numerous post-translational modifications on AR, including phosphorylation and lysine and arginine methylation[75,76], which contribute to disease pathogenesis[17,77–82]. Some of these post-translational modifications occur in the cytosol before or upon androgen binding, while others occur in the nucleus. LSD1 and PRMT6 localization is not restricted to the nucleus, and they may have different functions in the cytosol than in the nucleus. PRMT6-mediated arginine methylation of AR prevents phosphorylation by Akt and vice versa[20]. Because Akt-mediated phosphorylation of AR reduces androgen binding and enhances AR degradation by the ubiquitin-proteasome system[83], it is possible that arginine methylation in the cytosol is important for AR stabilization. Further, the role of PRMT6 is broader than expected in polyQ diseases. PRMT6-mediated methylation of huntingtin regulates the key function of axonal transport and is affected by polyQ expansion[84]. Less obvious is the functional role of LSD1/AR and LSD1/PRMT6 interactions in the cytosol. Perhaps AR and PRMT6 are targets of LSD1, and possibly lysine demethylation helps prepare AR or PRMT6 to carry out specific cytosolic functions.

A key finding here was that LSD1 and PRMT6 overexpression was an early pathological process specifically in skeletal muscle in a cell-autonomous fashion. Although it is not known whether PRMT6 exerts any physiological function in skeletal muscle, its expression is upregulated in pathological settings associated with muscle atrophy, including fasting and muscle-specific PRMT1 knock-out[85], as well as high-fat diet[25]. LSD1 expression is enhanced during skeletal muscle

differentiation and regeneration[49,86], and LSD1 regulates expression of myogenic genes, such as myogenin, and oxidative metabolism genes[48,50], suggesting a physiological role for LSD1 in muscle. Chronic LSD1 overexpression may potentially become toxic, as LSD1 toxic GOF contributes to and is a valuable therapeutic target for dilated cardiomyopathy[87], transverse aortic constriction[88], and SBMA, as reported here.

Gene silencing is a valuable therapeutic approach for neuro-degenerative diseases associated with toxic GOF mechanisms. Familial forms of amyotrophic lateral sclerosis with mutations in the gene superoxide dismutase 1 (SOD1) are now treated with Tofersen, an intrathecally administered antisense oligonucleotide against SOD1, showing encouraging results in patients[89]. Several proof-of concept studies have shown promising results in C9orf72 expansions, fused in sarcoma-linked amyotrophic lateral sclerosis, Huntington's disease and spinocerebellar ataxia type 1, 3, and 7 mouse models and non-human primates as well as in phase I clinical trials[90–96]. In the case of SBMA, males are hemizygous for AR, and silencing polyQ-expanded AR itself may enhance signs of androgen insensitivity in patients[7]. Here we provide evidence that targeting two overexpressed AR co-regulators that enhance mutant AR toxic GOF modifies expression of genes involved in pathways dysregulated in SBMA muscle and key to muscle phy-siology and homeostasis. These pathways include muscle con-traction and myofibril assembly, glycolysis, and metabolism. Although the number of rescued genes is limited with respect to the total number of genes dysregulated in SBMA muscle, this effect is associated with amelioration of the severe disease phe-notype in a mouse model of SBMA. A final consideration related to our preclinical approach was that, although the amiR was deliv-ered systemically, downregulation of LSD1 and PRMT6 was sig-nificant only in skeletal muscle, where these co-activators are overexpressed. LSD1 and PRMT6 are likely not the only AR co-regulators altered in SBMA tissues, and there is a need for further investigation of the impact of dysregulated co-regulators in vul-nerable SBMA tissues. This information may reveal how the interaction of AR with its co-regulators results in a coordinated and regulated cell type-specific gene expression program and may allow us to decipher how polyQ expansions affect these processes, resulting in altered gene expression in tissues that primarily degenerate (muscle and motor neurons). Also, it may allow us to determine how coexisting GOF and LOF processes cause the cen-tral and peripheral symptoms that manifest in patients.

## Methods

### Animals and treatments

Our research complies with all relevant ethical regulations. Animal care protocols conform with the appropriate national legislation (art. 31, D.lgs. 26/2014) and guidelines of the Council of the European Com-munities (2010/63/UE). This study was approved by local ethics com-mittees (Universities of Trento approval number 974/2020-PR, and Padova approval numbers 1289/2019-PR, 207/2020-PR) and the Italian Ministry of Health. The mice were pathogen free according to the FELASA list (FELASA 2014). Animals were housed in a single ventilated cage (Tecniplast Green Line Sealsafe PLUS Mouse) with autoclaved commercial soil bedding, food, and enrichment. Mice were fed with a certified rodent diet (SDS VRF1 (P)). Mice were monitored on a daily basis by specialized operators and by the designated veterinary. The colonies were monitored by a sentinel program. Mice were euthanized by either administration of carbon dioxide or by the mix of Alfaxalone (60 mg/kg) and Xylazine (10 mg/Kg).

AR100Q transgenic and AR113Q knock-in mice were genotyped using My Taq Extract-PCR (Bioline, #BIO-21127) following manu-facturer's instructions. Briefly, ears were punched and collected in a tube, mixed with Buffer A, Buffer B and water, placed at 75 °C for 5 min,

and finally at 95 °C for 10 min. The DNA was then amplified by using the MyTaq HS Red Mix, 2x and the human AR pair of primers (AR100Q mice: Forward 5'-CTTCTGGCGTGTGACCGGCG, reverse 5'-TGAGC TTGGCTGAATCTTCC; AR113Q mice: Forward 5' CCACGTTGTCCC TGCTGGGCCCCAC, reverse 5' GACACTGCTTTACACAACTCCTTGGC). The PCR product was run on 2% Agarose gel for AR100Q mice and on 3% Agarose gel for AR113Q mice in Tris-Acetate-EDTA (TAE) 1X Buffer at 80 V for 30 min[10,28]. Orchiectomy (surgical castration) was performed between 4 and 5 weeks of age. The testes were removed or left intact (Sham-operated)[97]. Mice were subjected to intraperitoneal injection of either saline solution or amiR-Prmt6/Lsd1 AAVs at 21 days of age and were evaluated weekly during weeks 4–14. Mice were euthanized when they lost >20% of body weight with respect to the highest weight measurement. For rotarod and hanging wire tasks, mice were rando-mized, and both genotype and AAV injection were disguised to the operator. Animals were trained to run on an accelerated rotarod (4–40 rpm) (Panlab, Harvard Apparatus, #LE8205) for a maximum of 300 s. Latency to fall was recorded, and the best performance of three trials was reported. For the hanging wire test, mice were placed on top of a wire cage lid, which was gently shaken three times to cause the mice to grip the wires, and then the lid was turned upside down. The latency to fall—for a maximum of 60 s—was recorded. For survival analysis, moribundity was the time in which the mouse lost 20% of body weight or showed inability to move, dehydration, and cachexia.

The research work done in flies has been reviewed and approved by the University of Pittsburgh Institutional biosafety office. All Dro-sophila stocks were maintained on standard cornmeal medium and fed 2 mM DHT at 28 °C in light/dark-controlled incubators. Lsd1 and Dart8 lines were obtained from the Vienna Drosophila Resource Center (VDRC, stock ID for DART8: v100228, and dLsd1: v106147). The androgen receptor (AR) line was generated by expressing human AR gene containing 52 polyglutamine repeats (AR52Q) in Drosophila[11]. Eye images were taken with a Leica M205C dissection microscope equip-ped with a Leica DFC450 camera and were processed using the Leica software. The quantification of external eye degenerative phenotype was done using criteria including ommatidial degeneration, brittle, loss of pigmentation, area of degenerated eyes and disorganization[11].

### Human samples

Deanonymized control (n = 5) and patient biopsy samples (n = 5) were obtained from the Neuromuscular Bank of Tissues and DNA Samples, Telethon Network of Genetic Biobanks, and EuroBioBank Network (Supplementary Table 1). This study was approved by the Ethics Committee for Clinical Practice of the Azienda Ospedale Università of Padova. All muscle biopsies were taken for diagnostic purposes after written informed consent was obtained from each patient according to the Helsinki Declaration. All patients who underwent muscle biopsy were clinically affected and showed weakness and/or fasciculation and/or muscle atrophy[98]. Myopathic changes together with neuro-genic atrophy were observed in muscle biopsies. Control samples were obtained from individuals with no neurological and neurodegenera-tive condition. SBMA lumbar spinal cord tissue was a gift from Dr. Lyle Ostrow (ALS Postmortem Tissue Core at Johns Hopkins University). Review Board approved the experiments using post-mortem tissue gifted by Dr. Ostrow and informed consent was obtained for all the samples in the present study. SBMA and control liver biopsy samples were collected[60]. The protocol for liver biopsies collection was approved by the NIH Intramural Combined Neuroscience IRB with protocol number 14-N-0099, and informed consent was obtained from all subjects.

### Vectors

Lentiviral constructs (pLKO.1-puro) for shRNA against PRMT6 (shPRMT6 #1 and shPRMT6 #2) and the corresponding scramble control were kindly provided by Ernesto Guccione. Guide RNAs were

cloned into lentiCRISPR v1 (Addgene, Plasmid #49535) using BsmBI restriction sites. *LSD1* and *PRMT6* oligos used for gRNA cloning are in Supplementary Table 2. HEK293T cells were transfected by calcium phosphate with lentiviral vectors together with pCMV-dR8.91 (Delta 8.9) plasmids containing gag, pol, and rev genes and VSV-G envelope plasmid. At 16 h post-transfection, medium was replaced with fresh medium, and 24 h later it was collected, centrifuged at $1000 \times g$ for 10 min (to pellet and thus remove any cellular debris), filtered through 0.45-µm pores, and stored at −80 °C in aliquots. To quantify viruses, 10 µL of viral particles were lysed for 10 min at RT by adding an equal volume of 2X lysis buffer [0.25% Triton X-100, 50 mM KCl, 100 mM Tris−HCl pH 7.4, 40% glycerol, and 0.8 U/µL RNase inhibitor (Ribo-Lock, Fermentas)]. Lysates were added to a single-step RT-PCR assay with 3.5 nM MS2 RNA (Roche) as template, 500 nM of each primer (5′-TCCTGCTCAACTTCCTGTCGAG-3′ and 5′-CACAGGTCAAACCTCC TAGGAATG-3′), and hot-start Taq (Truestart Hotstart Taq, Fermentas, #10540081), all in 20 mM Tris-Cl pH 8.3, 5 mM $(NH_4)_2SO_4$, 20 mM KCl, 5 mM $MgCl_2$, 0.1 mg/ml BSA, 1/20,000 SYBR Green I (Invitrogen, #S7563), and 200 µM dNTPs. SG-PERT reverse transcription assay was carried out with the following program: 42 °C for 20 min for reverse transcription reaction, 95 °C for 2 min for enzyme activation, followed by 40 cycles of denaturation at 95 °C for 5 s, annealing at 60 °C for 5 s, extension at 72 °C for 15 s, and acquisition at 80 °C for 5 s. A standard curve was obtained using known concentrations of high-titer viral supernatants (kindly provided by Dr. Massimo Pizzato, University of Trento, Italy). Lentiviruses were tested in vitro by transducing motor-neuron cell lines, and the most efficient knock-down was observed with shPRMT6 #1 (CACCGGCATTCTGAGCATCTT). Mutagenesis of LSD1-LXXAA mutant was performed by Vector Builder (https://en.vectorbuilder.com/).

### Cell cultures, transfection, and transduction

Cells were cultured in a humidified incubator with 5% $CO_2$ at 37 °C. MN1[30] and HEK293T (ATCC) cells were cultured and plated in Dulbecco's modified Eagle medium (DMEM, Euroclone #D642) [supplemented with 10% fetal bovine serum (FBS, Life Technologies, #10270106), 1% penicillin/streptomycin (pen/strep, Euroclone #ECB3001D), and 1% L-glutamine (Euroclone, #ECB3000D)]. LNCaP cells were cultured in RPMI complete medium [Gibco Roswell Park Memorial Institute (Euroclone, #ECB9006L) supplemented with 10% FBS, 1% pen/strep, and 1% L-glutamine]. C2C12 cells were cultured and differentiated to myotubes[31]. In brief, C2C12 cells stably expressing AR24Q and AR100Q were maintained in growth medium (DMEM supplemented with 10% FBS, 1% pen/strep, and 1% L-glutamine) and seeded at a density of 7000 cells/cm². Two days after seeding (day 0), growth medium was changed with the differentiation medium (DM) [DMEM supplemented with 2% horse serum (HS, Life Technologies, #16050122), 1% pen/ strep, and 1% L-glutamine] for 7 days. Every two days DM was changed and cells were treated with ethanol or DHT (10 nM, Sigma-Aldrich #10300). HEK293T cells were transfected with polyethylenimine (PEI, branched MW 25,000 Da, Sigma-Aldrich, #408727) according to well dimensions. DNA: PEI (0.5% v/v) ratio was 1:1. At 24 h after the transfection the DMEM complete medium supplemented with 10% FBS was changed to DMEM supplemented with charcoal stripped FBS (CDS) and cells were treated with vehicle (ethanol) or 10 nM DHT and harvested for the different analysis 24 h post-treatment. For the pharmacological inhibition, cells were treated with the LSD1 catalytic inhibitor tranylcypromine (TCP, Merck, #616431) and the PRMT inhibitor adenosine dialdehyde (AdOx, Sigma-Aldrich, #A7154), both at a final concentration of 10 µM, while the selective LSD1 inhibitor SP-2509 was used at a final concentration of 100 nm. MN1 cells were transfected with Lipofectamine 2000 according to manufacturer instructions (Thermo Fisher Scientific, #11668-019). MN1 and LNCaP cells were transiently transduced with lentiviruses (MOI 30) or transfected with 2 µg DNA using Lipofectamine 2000 CD (Thermo Fisher Scientific, #12566014). The following day, positively transfected cells were selected with 10 µg/µL Blasticidin (PanReacApplichem, #A3784,0025). After 24 h of selection or at 48 h post-transduction, cells were induced with 10 nM DHT in DMEM or RPMI medium supplemented with 10% CDS, 1% pen/strep, and 1% L-glutamine.

The SBMA iPSCs were derived and differentiated as previously reported[99]. iPSCs were maintained on Matrigel (Corning, #354277)-coated tissue culture dishes in E8 Flex medium (Thermo Fisher Scientific, #A2858501). The medium was changed every two days, and iPSCs were split every 4–6 days using Accutase (StemCell Technology, #07922). Culture medium was supplemented with 10 µM ROCK inhibitor (Tocris, #1254) on the day of passage. For differentiation, stably transfected iPSCs were induced with Neural Induction Medium containing 2 µg/mL doxycycline (#D9891) and 10 nM R1881. After 48 h of doxycycline treatment, motor neurons were dissociated with Accutase to single cells and re-plated on PDL/laminin-coated surfaces in Neural Differentiation Medium containing 2 µg/mL doxycycline and 10 nM R1881. On day 4, half of the cell culture medium was removed and replaced with Neuron Medium containing 10 nM R1881.

### Immunocytochemistry

For immunofluorescence, MN1 cells were fixed in 4% paraformaldehyde (PFA) for 20 min, washed with phosphate-buffered saline (PBS), and permeabilized in 0.1% Triton X-100 in PBS for 10 min. Cells were incubated with blocking buffer containing 10% normal goat serum (NGS) and 5% milk powder diluted in Tris-buffered saline (TBS) for 45 min. Cells were then incubated at room temperature for 2 h with the following primary antibodies: AR (H280, Santa Cruz, #sc-13062, 1:500), PRMT6 (Santa Cruz, #sc-55702, 1:500), and LSD1 (Abcam, #ab17721, 1:500). Secondary antibodies [Invitrogen (1:500): Donkey anti-Goat 647 (#A-21447), Donkey anti-Rat 555 (#A-78945); Donkey anti-Rabbit 488 (#A-21206); Donkey anti-mouse 405 (#A48257)] were incubated for 1 h in blocking buffer. Cells were then washed with TBS and incubated for 5 min with DAPI (Invitrogen) and mounted with Aqua-Poly/Mount (Polyscience Inc.). Images were acquired digitally with a NIKON Eclipse 80i upright microscope or with an inverted confocal microscope (Leica)[20]. For iPSC-derived MNs, 6 day-post-infection (dpi) immortalized motor neurons (iMNs) were washed with PBS and fixed with 4% paraformaldehyde (PFA) for 10 min at RT and washed again with PBS before antibody labeling. Cells were permeabilized with 0.1% Triton X-100 and 0.01% Tween-20 for 10 min at RT and then blocked with 10% BSA in 0.1% Triton X-100 and 0.1% Tween (PBST) for 1 h at RT. Samples were incubated overnight at 4 °C with primary antibodies in 3% BSA in PBST. After overnight incubation, samples were washed twice with PBS and incubated with fluorescent secondary antibodies (Invitrogen) in 3% BSA in PBST for 1 h at RT. Slides were mounted using ProLong Diamond Antifade Mountant with DAPI (Invitrogen, # D3571). For immunofluorescence analysis of post-mortem spinal cord tissue, frozen sections were fixed with 4% PFA and incubated overnight at 4 °C with primary antibodies. For immunofluorescence analysis the following antibodies were used: anti-PRMT6 (Santa Cruz, #sc-55702(Q16)) 1:50, anti-LSD1 (Abcam, #ab190507, 1:200), anti-HB9 (DSHB, #81.5C10, 1:100), and anti-AR (GeneTex, #GTX22742, 1:50; H280, Santa Cruz, #sc-13062, 1:100). Secondary antibodies [Invitrogen (1:500): Donkey anti-Goat 647 (#A-21447), Donkey anti-Rat 555 (#A-78945); Donkey anti-Rabbit 488 (#A-21206); Donkey anti-mouse 405 (#A48257)]. Digital images were captured with a Zeiss LSM 880 confocal microscope with a ×40 objective.

### Cell viability and proximity ligation assay (PLA)

MN1 AR24Q and AR100Q cells were plated in a 24-well plate at 50000 cells/well in DMEM. At 24 h post-transfection DHT was added to the medium, and on the following day the cells were processed for 3-[4,5-dimethylthiazol-2-yl]-2,5 diphenyl tetrazolium bromide (MTT)

assays[34,100]. Briefly, MTT was directly added to the medium in a 1:10 ratio (50 μL/well) and incubated with 5% $CO_2$ at 37 °C for 30–45 min until purple precipitates were visible. Medium was replaced with 200 μL of dimethyl sulfoxide to dissolve the formazan product, and the plate shook for 10 min until all precipitates were completely dissolved. The solution was transferred to a 96-well plate, and absorbance at 570 nM and 690 nM was quantified on a Tecan Infinite 200 PRO spectrophotometer. Final absorbance was obtained by subtracting the signal at 690 nM to the signal at 570 nM. For proximity ligation assay (PLA), cells were fixed with 4% PFA for 20 min, washed three times with 1X PBS, and permeabilized with 0.1% Triton X-100 in PBS for 5 min. PLA analysis ([Duolink In Situ Red Starter Kit Mouse/Rabbit, Merck, #DUO92101]) was performed by following manufacturer's protocol as previously described[101]. In brief, cells were blocked using Duolink® Blocking Solution and incubated in a heated humidity chamber for 1 h at 37 °C. The blocking solution was then removed and the primary antibody—diluted in the Duolink® Antibody Diluent—was directly added to the slide and incubated overnight at 4 °C. The following primary antibodies used were incubated overnight at 4 °C: anti-LSD1 (Abcam, #ab17721, 1:2000), anti-AR (AR 441, Santa Cruz Biotechnology, #sc-7305, 1:2000), anti-PRMT6 (Bethyl, #A300-929A, 1:2000), and anti-PRMT6 (Abcam, #ab151191, 1:2000). The day after, the primary antibody solution was removed, the slides were washed twice for 5 min in 1X Wash Buffer A at room temperature, and the PLA probe solution was added. The slides were incubated in a pre-heated humidity chamber for 1 h at 37 °C. After the incubation, the solution was removed, slides were washed twice, and the ligation solution was applied. Slides were incubated in a pre-heated humidity chamber for 1 h at 37 °C. Finally, slides were washed again, and amplification solution was added. Slides were incubated—always in a pre-heated humidity chamber—for 100 min at 37 °C. The cells were washed twice in 1X Wash Buffer B for 10 min at room temperature and in 0.01X Wash Buffer B for 1 min. Slides were mounted with a coverslip using a minimal volume of Duolink® In Situ Mounting Medium with DAPI and Slides were imaged with a ×63 oil immersion objective using the Zeiss Axio Observer Z1 inverted microscope, and PLA positive red dots were quantified using ImageJ 1.51 software.

### EdU staining
LNCaP cells were plated on coverslips pre-treated with poly-D-Lysine (Merck, #P7280) in a 12-well plate at 160000 cells/well in RPMI. At 24 h post-transfection positively transfected cells were selected with 10 μg/μL blasticidin (PanReacApplichem, #A3784,0025). DHT was added to the medium, and the following day EdU assay was performed (Click-iT Plus EdU Alexa Fluor 594 Imaging Kits, Invitrogen, #C10639). For the assay, cells were incubated with 10 μM EdU for 30 min. The medium was changed with fresh RPMI, and after 10 min cells were washed with PBS pre-warmed at 37 °C. Cells were fixed with cold 4% PFA for 10 min, washed twice with 3% BSA in PBS for 5 min, and permeabilized with 0.5% Triton X-100 in PBS for 20 min. Cells were washed twice, and Click-iT Plus reaction cocktail was added to each coverslip for 30 min at RT in the dark. Cells were washed once with 3% BSA in PBS and with PBS for 5 min. Hoechst 33342 (Thermo Fisher Scientific, #62249, Component G, 5 μg/mL) was added for 30 min at RT in the dark. Cells were washed twice with PBS, and coverslips were mounted on slides. Slides were imaged with a ×20 objective using the Zeiss Axio Observer Z1 inverted microscope, and EdU-positive red cells were quantified using ImageJ 1.51 software.

### Quantitative real-time PCR
Total RNA was extracted with TRIzol (Thermo Fisher Scientific, #15596018), and RNA was reverse transcribed using Superscript Reverse Transcriptase III (Invitrogen, #18080093) following the manufacturer's instructions. Gene expression was measured by RT-qPCR using SsoAdvanced Universal Sybr green supermix (Bio-Rad,

#1725274) and the QuantStudio™ 5 Real-Time PCR System (Thermo Fisher Scientific). Gene expression was normalized to *actin* expression levels. The complete list of primer sequences (Eurofins) is provided in Supplementary Table 4. For RT-PCR in iPSC-derived motor neurons, 500 ng of RNA was reverse transcribed with the cDNA Reverse Transcription kit (Invitrogen, #4368814) and qPCR was performed in triplicate wells per sample using the Taqman Gene Expression Master Mix (Thermo Fisher Scientific, #4369510) on the QuantStudio 6 Flex machine (Thermo Fisher Scientific). The Taqman assays used: HPRT1 (Hs02800695), PRMT6 (Hs00250803), and LSD1 (Hs01002741).

### RNA-seq
RNA was extracted from samples with TRIzol, following the manufacturer's protocol. RNA was quantified using Nanodrop and Qubit, and quality was assessed on an Agilent 2100 Bioanalyzer. Purified RNA served as the input for cDNA library preparation with TruSeq Stranded mRNA (Illumina) according to the manufacturer's protocol. cDNA library fragment size was determined by BioAnalyzer 2100 HS DNA assay (Agilent). Libraries were sequenced as paired-end reads on a NovaSeq6000 (Illumina).

### Computational analysis
For RNA-seq, data for amiR-*Lsd1*/*Prmt6*-treated AR100Q mice, AR100Q mice, and WT mice comparisons were analyzed by Rosalind (https://rosalind.onramp.bio/), with a HyperScale architecture developed by Rosalind Inc. (San Diego, CA), while data for WT mice and amiR-*Lsd1*/*Prmt6*-treated WT mice were analyzed on local computational resources. In both cases, reads were trimmed using cutadapt[102], quality scores were assessed using FastQC and reads were aligned to the *Mus musculus* genome (mm10) using STAR[103]. Individual sample reads were quantified using HTseq and normalized via relative log expression using DEseq2[104], which also was used for differential expression analyses. *P*-values were adjusted for multiple hypothesis testing using the Benjamini and Hochberg method. Differentially expressed genes between non-treated (AR100Q) and treated (amiR-*Lsd1*/*Prmt6*) SBMA mice and WT controls were those genes with absolute fold-change >4 and adjusted *p* < 0.01. Differentially expressed genes between WT and treated WT were instead searched using the less stringent absolute fold-change >2 and adjusted *p* < 0.01. We defined a set of "rescued genes" as genes showing a significant but opposite direction in AR100Q vs WT and amiR-*Lsd1*/*Prmt6* vs AR100Q differential expression analyses. Additionally, we identified a subset of rescued genes, referred to as "totally rescued," showing no significant differential expression between amiR-*Lsd1*/*Prmt6* and WT. The difference between rescued and totally rescued genes were "partially rescued" genes. Functional enrichment analysis of rescued genes was performed using Metascape[105], using the following gene lists: GO biological processes, KEGG pathways, and Reactome gene sets. All genes in the genome were used as enrichment background. Terms with *p* < 0.01, minimum count of 3, and an enrichment factor >1.5 were collected and grouped into clusters based on membership similarities. The most statistically significant term within each cluster was chosen to represent the cluster. When comparing gene lists A and B using the overlapping coefficient measure, cardinality of the intersection between A and B was divided by minimum cardinality between cardinalities of A and B sets. DEGS are in Supplementary Data 1.

### Biochemistry
Cells were lysed for western blot analysis in RIPA buffer (6 mM $Na_2HPO_4$, 150 mM NaCl, 4 mM $NaH_2PO_4$, 150 mM NaCl, 2 mM EDTA pH 8.0, 1% sodium deoxycholate, 0.5% Triton X-100) plus fresh protease inhibitors (Sigma-Aldrich, #P8340). Samples were incubated 20 min on ice and then centrifuged for 15 min at 21,000 × *g* at 4 °C. Supernatant was collected and stored at −80 °C or further processed for western blotting.

Frozen tissues were pulverized using a mortar and pestle on dry ice, transferred to a cold sample tube, and resuspended in 2% SDS-RIPA buffer (50 mM Tris–HCl pH 8.0, 150 mM NaCl, 1% NP40, 0.5% sodium deoxycholate, 2% SDS) with fresh protease inhibitors (Sigma-Aldrich, #P8340). Lysates were sonicated and centrifuged at $21,000 \times g$ for 15 min at 4 °C.

Protein concentration was measured using BCA assay (Pierce BCA Protein Assay, Thermo Fisher Scientific). For western blotting, equal amounts of protein extracts from tissues or cell lysates were boiled at 95 °C for 5 min in 5X sample buffer (62.5 mM Tris–HCl pH 6.8, 2% SDS, 25% glycerol, 0.05% bromophenol blue, 5% β-mercaptoethanol) and separated by SDS-PAGE. Proteins were transferred to 0.45-mm nitrocellulose membranes (Bio-Rad, #162-0115), blocked for 1 h in 5% non-fat milk/BSA in TBS buffer/0.1% Tween, and incubated with primary antibodies (see below) for 2 h at RT or overnight at 4 °C. HRP-conjugated secondary antibodies were incubated for 1 h at RT (1:5000 dilution in blocking solution, Bio-Rad, goat anti-rabbit #1706515, goat anti-mouse #1706516), and signals were detected with Chemidoc (Bio-Rad) or Alliance Q9 Mini chemidoc system (Uvitec, Cambridge, UK).

For immunoprecipitation assays, cultured cells were washed twice with ice-cold PBS, scraped, collected, and spun at $1200 \times g$ for 3 min at 4 °C. Pellets were lysed in IP buffer (50 mM HEPES, 250 mM NaCl, 5 mM EDTA, 0.1% NP40) containing 1 mM PMSF and fresh protease inhibitors. Cell pellets were homogenized by passing through syringes (2.5 mL-22Gx1¼″ and 1 mL-25Gx5/8″), incubated on ice for 45 min, and centrifuged at $21,000 \times g$ for 30 min at 4 °C. Supernatant was transferred and quantified by Pierce BCA Protein Assay (Thermo Fisher Scientific). Protein extract (1.5–4 mg) was incubated with primary antibodies (see below) overnight at 4 °C with rotation. Complexes were incubated with protein A/G plus-agarose (Santa Cruz Biotechnology, #sc-2003) beads for 2 h at 4 °C with rotation. Antigen–antibody complexes were washed three times with lysis buffer and once with wash buffer. Bound protein was eluted with 35 µL of 2X SDS buffer and denatured at 95 °C for 5 min before loading on SDS-PAGE.

For immunoprecipitation assays in tissue, quadriceps muscles were lysed in IP buffer (50 mM HEPES, 250 mM NaCl, 5 mM EDTA, 0.1% NP40) with fresh protease inhibitors (Sigma-Aldrich, P8340), incubated on ice for 30 min, and centrifuged at $21,000 \times g$ for 45 min at 4 °C. Supernatant was transferred and quantified by Pierce BCA Protein Assay (Thermo Fisher Scientific). Protein extract (2 mg) was pre-cleared with pre-washed Pierce Protein A Magnetic Beads (30 µL, Thermo Fisher Scientific, #88845) for 1 h at 4 °C. Beads were discarded, and the pre-cleared lysate was incubated with 2 µg of primary antibody (PRMT6, Bethyl, #A300-929A) or nonimmune rabbit immunoglobulin G (IgG) overnight at 4 °C with rotation. The following day, antigen–antibody complexes were incubated with new pre-washed Pierce Protein A Magnetic Beads (20 µL, Thermo Fisher Scientific, #88845) for 2 h at 4 °C with rotation. Antigen–antibody complexes were washed four times with lysis buffer. Bound protein was eluted with 30 µL of NuPAGE LDS Sample Buffer (4X) and 0.1 M DTT, heated at 70 °C for 5 min, and loaded onto 10% SDS-PAGE gels.

The following antibodies were used for analysis: anti-LSD1 (Abcam, #ab17721, 1:1000), anti-PRMT6 (Proteintech, #15395-1-AP, 1:1000; Bethyl, #A300-929A, 1:2000), anti-AR (441, Santa Cruz, #sc-7305; H-280, Santa Cruz, #sc-13062, 1:1000), anti-FLAG (Sigma-Aldrich, #7425, 1:1000), anti-GFP (Roche, #11814460001, 1:1000), anti-calnexin (Enzo, #ADI-SPA-860, 1:2500), and anti-β-tubulin (Sigma-Aldrich, #T7816, 1:10,000). Quantifications were performed using ImageJ 1.51 software.

Nicotinamide adenine dinucleotide (NADH) staining[16] was performed on slides with flash-frozen in liquid nitrogen skeletal muscles previously embedded in optimal cutting temperature (OCT) compound (Qpath, #411243), and cut in 10 µm thick-cross sections with a cryostat (Leica, #CM1850UV). Slides were immersed in fresh NADH solution [(5% Nitrotetrazolium Blue Chloride (Sigma-Aldrich, #N6876),

4% NADH (Sigma-Aldrich, #N8129) dissolved in 50% Buffer Tris–HCl 0.2 M pH 7.4)] for 20 min and protected from light. When purple color appeared on muscles, slides were quickly washed with acetone solutions at different percentages: 30%, 60%, 90%, 60%, 30%. Finally, slides were washed twice in water and let them dry in vertical position. Once dried, slides were mounted with Eukitt® (Sigma, #03989). Images were taken using an upright epifluorescence microscope (Zeiss Axio Imager M2) equipped with a Vis-LED light source and a Coloured Camera (AxioCam MRc, 5 Megapixel; pixel size: 6.45 µm × 6.45 µm) with an EC Plan-Neofluar 10×/0.3 objective. Multichannel images were taken using Zeiss Axio Vision Software (V.4.8) and the number of oxidative and glycolytic fibers was counted for each mosaic.

Chromatin immunoprecipitation AR (ChIP) was performed using $4.8 \times 10^7$ C2C12 myoblasts expressing either AR24Q or AR100Q[32]. Cells were treated with 10 nM DHT 24 h after transfection. At 48 h, cells were cross-linked with 1% formaldehyde for 10 min and harvested and lysed. Cell lysates were sonicated to an average DNA fragment length of 200 to 400 bp. The experiments were conducted using 300 µg of chromatin. AR was immunoprecipitated with anti-AR antibody (Millipore, #06-680, 14 µg) or anti-mouse immunoglobulin G (IgG) antibody (Cell Signaling Technology, #5415s). ChIP list is provided in Supplementary Table 5).

For histone purification, cells were seeded at a density of 7000 cells/cm² and cultured in 10% FBS. When they reached 70–80% confluence, cells were grown in differentiation medium differentiation medium (DM) [DMEM supplemented with 2% horse serum (HS, Life Technologies, #16050122), 1% pen/strep, and 1% L-glutamine] at 37 °C in a humidified atmosphere containing 5% $CO_2$. Cells were replenished every three days. Histones were extracted from myocytes by acid extraction[106]. Briefly, cells were washed twice with ice-cold PBS and lysed in 1 mL ($5 \times 10^6$ cells/mL) hypotonic lysis buffer (10 mM Tris–HCl pH 8.0, 1 mM KCl, 1.5 mM $MgCl_2$, 1 mM DTT, 0.5 mM PMSF, 1X protease and phosphatase inhibitor) and incubated for 30 min with rotation at 4 °C. Nuclei were resuspended in 400 µL 0.4 N $H_2SO_4$ and incubated at 4 °C for 30 min. Supernatant containing histones was precipitated with 132 µL of TCA at 4 °C overnight. Following centrifugation, the pellet was washed twice with 1 mL of ice-cold acetone and air-dried. Histones were suspended in water and stored at −80 °C. Histones were quantified by colorimetric DC protein assay (Bio-Rad, #5000111) according to the manufacturer's instructions, with 5–10 µg of protein used for further analysis. The following antibodies used were: anti-H3K3me2 (Abcam, #ab7766, 1:1000) and anti-H3 (Abcam, #ab1791, 1:1000).

For luciferase assays, HEK293T cells were transfected with vectors expressing non-expanded and polyglutamine-expanded AR together with vectors expressing the luciferase reporter gene under control of a canonical ARE (ARE-Luc)[20]. To normalize data for transfection efficiency, cells expressing the Renilla reporter gene under control of timidine kinase (TK-Ren) were transfected with ARE-Luc in a ratio of 1:10. Cells were treated with vehicle or DHT for 16 h and then processed for luciferase assay, following the manufacturer's instructions (Promega, #E1910).

### Generation of amiRs targeting LSD1 and PRMT6

amiRs targeting either *PRMT6* or *LSD1* in mouse and human cells were designed with the online tool BLOCK-iT RNAi Designer (Supplementary Table 3). amiRs were inserted into pcDNA6.2-GW/EmGFP-miR plasmid using the BLOCK-iT Pol II miR RNAi kit (Invitrogen, #K493600) and carrying a spectinomycin resistance cassette[107]. The expression cassette consisted of a 5′ miR flanking region, target-specific stem-loop amiR sequence, and 3′ miR flanking region. This amiR cassette can be expressed from the 3′ untranslated region of any reporter gene under the control of an RNA polymerase type II promoter[108]. As a negative control, we used the control amiR sequence from pcDNA6.2-GW/EmGFP-miR-neg-control plasmid (provided with the kit) containing a

sequence that does not target any known vertebrate gene (control amiR: AAATGTACTGCGCGTGGAGAC). Top-10 competent *E. coli* were transformed, and positive clones were selected using EGFP forward primer 5′-GTCCTGCTGGAGTTCGTG-3′. amiR sequences against *PRMT6* (#1: TGCTGAATCAGACCATGTTGCCTTTCGTTTTGGCCACTGA CTGACGAAAGGCAATGGTCTGATT) and *LSD1* (#2: TGCTGTTGATGAG AGGTATACATCACGTTTTGGCCACTGACTGACGTGATGTACCTCTCAT CAA) were sub-cloned downstream of EGFP under control of the CAG promoter in an AAV vector derived from pAAV-CAG-EGFP (Addgene, #37825)[107].

### Viral production and titration

AAV serotype 9 (AAV9) was produced using a slight modification of the adenovirus-free transient transfection method[109]. Briefly, adherent HEK293 cells grown in roller bottles were transfected with three plasmids containing adenovirus helper proteins, AAV Rep and Cap genes, and the ITR-flanked transgene expression cassette. Three days after transfection, cells were harvested, lysed by sonication, and treated with benzonase (Merck-Millipore, #101697). Vectors were purified using two successive ultracentrifugation rounds in cesium chloride density gradients. Full capsids were collected. The final product was formulated in sterile PBS containing 0.001% of pluronic F-68 (Sigma-Aldrich, #P1300) and stored at −80 °C. Titers of AAV vector were as follows: control amiR: 3.8E12 vg/mL; amiR-*Prmt6*#6: 4.18E13 vg/mL; amiR-*Lsd1*#1: 9.8E12 vg/mL; and amiR-*Prmt6*/*Lsd1*: 3.3E13 vg/mL.

### Statistical analysis

Statistical analysis was performed with Jamovi (Version 2.3.18.0) or Microsoft Office Excel (Microsoft, version 2017). When needed, normality was tested using the Kolmogorov-Smirnov goodness-of-fit test. To compare the mean difference of a dependent variable between independent groups, two-sample t-tests and one-way analysis of variance (ANOVA), were used for two and more than two groups, respectively. Two-way ANOVAs were performed to evaluate the effects of two categorical predictors on a dependent variable. For all ANOVAs, follow-up post hoc tests were conducted for pairwise comparisons. All tests were two-tailed and the significance threshold was set at $p < 0.05$.

### Reporting summary

Further information on research design is available in the Nature Portfolio Reporting Summary linked to this article.

## Data availability

All data generated or analyzed during this study are included in this article and its Supplementary Information files. All requests for raw data and materials should be addressed to the corresponding author. Any data and materials that can be shared will be released via a material transfer agreement. Raw RNAseq FASTQ files and counts data for non-treated (AR100Q), treated (amiR-Lsd1/Prmt6) SBMA mouse models and WT controls have been deposited in the Gene Expression Omnibus (GEO) database under accession code GSE193539. Source data are provided with this paper.

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

## Acknowledgements

We thank Dr. Kenneth Fischbeck (NINDS, NIH) for insightful comments and suggestions. We also thank Morena Simonato (CNR, Italy) for technical support, Dr. Sergio Robbiati at the Model Organism Facility, and Dr. Giorgina Scarduelli at the Advanced Imaging Facility at Dept. CIBIO, University of Trento for technical help. We would like to thank Veronica De Sanctis and Roberto Bertorelli at the NGS Facility at Dept. CIBIO, University of Trento for the RNA sequencing experiments and the CRIBI, University of Padova, for the cDNA library preparation. We thank Dr. Lyle Ostrow, and the ALS Postmortem Tissue Core for providing the postmortem tissue. This work was supported by Dulbecco-Telethon Institute/Fondazione Telethon-Italy (TCP12013 and GGP19128 to M.P.), Association Française contre les Myopathies (AFM-20658 to M.B., and AFM-22221 to M.P. and M.B.), Kennedy's Disease Association (to L.T., E.Z., and M.B.), NIH-R21 (1R21NS111768-01 to M.P. and U.B.P.), Fondazione AIRC-Italy (24423 to M.P.), Program "Rare diseases" CNCCS Scarl Pomezia (to M.P.), U.S. National Institutes of Health (1R21NS111768 to M.P. and U.B.P.), Fondazione Caritro (to A.M. and L.T.).

## Author contributions

R.P. performed analysis of AR/LSD1/PRMT6 interaction in cells, transcriptional assays, and biochemical analyses in cells; A.B. performed analysis in mice, prostate cancer cells, PLA, biochemical analyses in cells and tissues; R.A. analyzed AR transactivation in cells and molecular and biochemical analyses in cells and mouse tissues; D.T. cloned and selected the amiR in AAV vector;  E.Z. and C.M. performed transcript and protein analysis in cells and tissues; D.D. and A.R. performed computational analysis of transcriptomic data; G.P. and A.C. designed CRIPSR guides; E.N.A. and U.B.P. performed experiments in flies; A.M. and L.T. helped with the cloning strategies; E.B. provided constructs for LSD1 expression and data interpretation; G.S. provided analysis of patient clinical assessment; W.F.L. and C.R. performed CHIP assays; F.S. performed statistical analysis; N.P. and C.G. performed experiments in iPSCs and liver biopsies; A.C. and G.R. contributed to amiR design, in silico validation, and biodistribution; M.B. and M.P. conceived the study, provided main financial support, analyzed and interpreted data, and wrote the manuscript.

## Competing interests

M.P., M.B., G.R., and A.C. are named as co-inventors on the patent application Italian Priority N. 102022000026595 "New inhibitors of epigenetic regulators/nuovi inibitori di regolatori epigenetici". The other authors declare no competing interests.

## Additional information

[1]Dulbecco Telethon Institute at the Department of Cellular, Computational and Integrative Biology, University of Trento, Trento, Italy. [2]Department of Cellular, Computational and Integrative Biology, University of Trento, Trento, Italy. [3]Department of Biomedical Sciences, University of Padova, Padova, Italy. [4]Veneto Institute of Molecular Medicine, Padova, Italy. [5]Padova Neuroscience Center, Padova, Italy. [6]Department of Pediatrics, Children's Hospital of Pittsburgh, University of Pittsburgh Medical Center, Pittsburgh, PA, USA. [7]Department of Medical Biotechnology and Translational Medicine, University of Milan, Milan, Italy. [8]Department of Neuroscience, University of Padova, Padova, Italy. [9]MDUK Oxford Neuromuscular Centre, University of Oxford, Oxford, UK. [10]Institute of Developmental and Regenerative Medicine, Department of Paediatrics, University of Oxford, Oxford, UK. [11]National Institute of Neurological Disorders and Stroke, National Institutes of Health, Bethesda, MD, USA. [12]Istituto Italiano di Tecnologia, Genoa, Italy. [13]Université Paris-Saclay, Univ Evry, Inserm, Genethon, Evry, France. [14]Genethon, 91000 Evry, France. [15]Present address: Medical Research Council Laboratory of Molecular Biology, Cambridge, UK. [16]Present address: Wellcome Sanger Institute, Wellcome Trust Genome Campus, Hinxton, Saffron Walden, UK. [17]These authors contributed equally: Ramachandran Prakasam, Angela Bonadiman, Roberta Andreotti.
✉e-mail: manuela.basso@unitn.it; maria.pennuto@unipd.it

