## [Peer Review File · Nature Communications]

LSD1/PRMT6-targeting gene therapy to attenuate androgen receptor toxic gain-of-function ameliorates spinobulbar muscular atrophy phenotypes in flies and miceREVIEWER COMMENTS

Reviewer #1 (Remarks to the Author):

This is a well written manuscript reporting an important and exciting finding that has significant clinical relevance. The disease of interest is an androgen-dependent, male biased neuromuscular disease called SBMA. While it is now understood that this disease depends on both a polyglutamine expansion mutation in the androgen receptor (AR) and male levels of androgen, still too little is known about how these two factors work together to cause the disease and like so many other neurodegenerative diseases, SBMA still awaits a cure or even a treatment that would effectively counter the broad and devastating effects of this disease on motor function. The reported work is significant because it suggests a novel therapeutic strategy that has the potential to selectively reverse the effects of AR toxicity at its source. I applaud this team for their comprehensive comparison and analysis of the published evidence for gene dysfunction in diseased muscle that looks beyond their own model. I have only a few comments for the authors to consider that could improve the manuscript.

1. The authors propose that reducing the function of coregulators PRMT6 and LSD1 can reduce the toxic gain of AR function without loss of normal AR function. The evidence presented that reducing PRMT6/LSD1 function selectively reduces toxic gain of function AR function without disrupting normal AR function is not compelling (pg 12: 17% vs. 20%). I suggest removing this assertion from the Discussion (pg 13). However, I do encourage the authors to discuss this important issue, since any effective SBMA therapeutic will likely have to begin at puberty, when normal AR function must be spared to allow for the full expression of male traits.

2. While studying a range of models often adds strength to a scientific report, in this case, added strength is much less clear. For example, there is no justification for why some models were chosen over others. Few of the mechanistic studies were conducted using cultured myotubes, despite the clear focus on skeletal muscles in other ways. Why were human stem cells differentiated into motoneurons and not muscle cells? This is particularly important in the context of steroid receptor coregulators that can have, indeed do have, cell specific effects. This is not a trivial issue and should be addressed in the Discussion, including an explanation for why different models were chosen and also acknowledging how studies on one cell type may not provide insight into the mechanisms occurring in another cell type.

3. While over-expression of LSD1/PRMT6 is shown to be androgen-dependent in muscle (Fig 1), in line with the androgen-dependent nature of this disease, the in vitro evidence that each factor complexes with AR is androgen-independent (Pg 6), presumably unexpected. This finding is problematic in light of the androgen-dependence of the disease and also how and when coregulators are recruited to the AR (also typically a ligand-dependent event). This important issue should be addressed in the Discussion.

4. While the authors make a point of including a positive control for their PLA studies, they do not include negative controls controlling for the possibility that the different primary antisera cross-react with the target proteins. Please include these essential negative controls.

5. The authors assert that denervation is a key aspect of SBMA pathology, listing several proteins that are upregulated after denervation. However, what the authors do not mention is that many such proteins are also increased when the system is less active, i.e. expression of such proteins is activity-dependent, as typically shown using α -bungarotoxin or curare to silence muscles. Evidence is poor that SBMA muscles are physically denervated, even in late-stage diseased mice, but there is compelling evidence for functional denervation, as recent published reports demonstrate. Unless anatomical denervation has been shown in the AR100 model, this issue deserves a more nuanced presentation.

6. In vivo expression of coregulators was shown to be androgen-dependent, raising the important question of whether the genes for LSD1 and/or PRMT6 have AREs. Such studies would add strength to the paper but at the least, this question should be discussed in the Discussion.

Reviewer #2 (Remarks to the Author):

Review Of Nature Communication 3552140

LSD1/PRMT6-targeting gene therapy to attenuate androgen receptor toxic gain-of-function ameliorates SBMA phenotype

SBMA is caused by the expansion of a polyglutamine repeat in the androgen receptor (AR) coding region. The current study is to use Crispr/Cas9 targeting the androgen receptor (AR) co-activators (LSD1/PRMT6) to treat the SBMA. The authors used a variety of cellular and animal models to show that 1) the LSD1 and PRMT6 are co-activators for the AR and that the co-activator expression levels are significantly increased in the presence of the mutant AR (AR100). 2) Crispr/Cas9 knock down the co-activators in the presence of mutant AR can rescue the phenotype in the muscle of transgenic SNMA mice and fly. Since clinical phenotypes of SBMA are featured by motor neuron degeneration and muscle weakness, the authors also used motor neuron derived cell lines and muscle cell lines.

Crispr/Cas9 has been used in a number of studies to test the treatment of rare diseases. The authors also sought to use this method to treat the SBMA animal models. Because targeting the mutant AR will also cause loss of function of the wildtype AR, the authors wanted to target the co-activators of the AR to achieve more selective therapeutic effects. Although the findings are interesting and developing an efficient treatment for the inherited neurodegenerative disease is urgently needed, there are some concerns that need to be addressed by additional experiments or at least clarified by discussion.

The present study has been done mostly in overexpression system, such as transgenic AR100 mice, transfected 293T cells and MN1 cells as well as transgenic flies. The use of over-expression and in vitro system in most of experiments in this study may be difficult to avoid artificial influences of overexpressed mutant AR on the phenotypes. The authors claim that they have AR113Q KI mice, muscle tissues from SBMA patients, and iPSC from SBMA but they did not use these samples for most of their critical experiments.

- 1). The authors claimed that pharmacology inhibition of AR interaction with co-factors causes loss of function of AR but they did not explain why targeted disruption would not have the same effect?
- 2). The authors claimed that the co-activators are in the same complex with AR based on the data of PLA staining. Since they have the specific antibodies, the results would be much stronger to use a co-IP study of SMBA knock-in mice to support this conclusion.
- 3). Increased LSD1/PRMT6 mRNA in muscle could result from muscle degeneration. In SMBA, motor neuronal degeneration is the primary cause of muscle degeneration. The current manuscript focused on LSD1/PRMT6 up-regulation in muscle without experiments to explain why these co-factors were upregulated in the SMBA models. In figure 1, most data are RT-PCR with the limited Western blotting result in Fig. 1B whose quantification does not seem to be consistent with RT-PCR results. Have the authors examined the expression of LSD1/PRMT6 proteins in the brainstem?

Minor concerns.

- 1). Figure 1b western blot should show all the samples
- 2). Please describe how the transactivation assay was done.
- 3). Western blots should show molecular weight.

Reviewer #3 (Remarks to the Author):

Spinobulbar muscular atrophy (SBMA) is caused by a CAG repeat expansion in the AR gene. PolyQ-expanded AR causes SBMA via gain-of-function(GOF) mechanism but partial loss-of-function(LOF) causes additional phenotypes, making therapeutic interventions complicated. Interaction of AR with its transcriptional co-regulators is necessary for GOF toxicity, however loss of these interactions may enhance the LOF SBMA phenotypes. Therefore, the authors here set out to identify AR co-regulators which are overexpressed in a region-specific manner and found LSD1 and PRMT6 overexpressed in skeletal muscle in patients and mice with SBMA. Pharmacologic and genetic manipulation of these factors and their interactions with AR attenuate SBMA in flies and mice.

The authors do a thorough job of demonstrating overexpression of Lsd1 and Prmt6 specifically in the skeletal muscle but not in other tissues in various SBMA mouse models, however, in human data they only show that these genes are overexpressed in skeletal muscle. Further data are needed in non-skeletal human tissue or cells to demonstrate that this phenomenon occurs in humans in addition to mice. The authors convincingly show that LSD1, like PRMT6, is a co-activator of polyQ-expanded AR and that by manipulating these interactions, AR transactivation can be decreased.

Next, the authors test a miRNA knockdown strategy in AR24Q and AR100Q mice, using an AAV9 vector containing miRNAs for both Lsd1 and Prmt6. The data convincingly show that significant knockdown of both LSD1 and PRMT6 lead to improvements in behavioral SBMA readouts, however data are lacking regarding potential negative effects of systemic Lsd1 and Prmt6 knockdown, and more transcriptional analyses are needed to explain why just a 10% rescue is seen.

There are a few things that the authors need to do to wrap up this interesting story:

1. Demonstrate in human non-skeletal cells or tissue that Prmt6 and Lsd1 are not overexpressed to truly conclude that in humans, like mice, these factors are overexpressed in a tissue-specific manner.
2. Demonstrate that systemic knockdown of Lsd1 and Prmt6 is not toxic. This is the primary concern for the paper, since the premise is that this method corrects toxic GOF without causing any toxic LOF, this needs to be further supported with additional data.
 - a. Assess more regions such as heart, lungs, adipose tissue, etc to test if PRMT6 and LSD1 protein levels are decreased upon IP injection with AAV9 amir-Lsd1/Prmt6.
 - b. Assess expression levels of AR targets in non-skeletal tissue to determine if levels change with PRMT6 and LSD1 reduction.
 - c. Assess expression levels of PRMT6 complex proteins and their known targets as PRMT6 is a general co-regulator and not only an AR co-activator.
3. The rescue of 10% of RNA-seq changes is very modest and the comparison of genes rescued in this model to public data sets in other SBMA models is weak.
 - a. Conducting RNA-seq in the WT mice treated with amir-Lsd1/Prmt6 group would be beneficial to determine if the low rescue is due to negative effects of miRNA masking the rescue or it is simply due to modest improvements in the transcriptome.
 - b. It would be nice to see knockdown of these co-regulators in an SBMA human cell model with transcriptional analyses conducted to test if human cells show rescue of SBMA transcriptional changes with Lsd1 and Prmt6 knockdown in addition to mice
4. Add WT CTR treatment group to weight figure in 8A.

5. WT amir-Lsd1/Prmt6 genotype could be moved from supplemental figure 7 to main figure 8A for weight and rotarod so it is clear that the amir-Lsd1/Prmt6 treatment does not cause a deficit independent of disease.

6. There is a mistake in supplemental figure 7 – the legend states this figure should be weight and rotarod but the graphs show hanging wire and rotarod.

Point-by-point response to the reviewers' comments

Reviewer #1 (Remarks to the Author):

This is a well written manuscript reporting an important and exciting finding that has significant clinical relevance. The disease of interest is an androgen-dependent, male biased neuromuscular disease called SBMA. While it is now understood that this disease depends on both a polyglutamine expansion mutation in the androgen receptor (AR) and male levels of androgen, still too little is known about how these two factors work together to cause the disease and like so many other neurodegenerative diseases, SBMA still awaits a cure or even a treatment that would effectively counter the broad and devastating effects of this disease on motor function. The reported work is significant because it suggests a novel therapeutic strategy that has the potential to selectively reverse the effects of AR toxicity at its source. I applaud this team for their comprehensive comparison and analysis of the published evidence for gene dysfunction in diseased muscle that looks beyond their own model. I have only a few comments for the authors to consider that could improve the manuscript.

1. The authors propose that reducing the function of coregulators PRMT6 and LSD1 can reduce the toxic gain of AR function without loss of normal AR function. The evidence presented that reducing PRMT6/LSD1 function selectively reduces toxic gain of function AR function without disrupting normal AR function is not compelling (pg 12: 17% vs. 20%). I suggest removing this assertion from the Discussion (pg 13). However, I do encourage the authors to discuss this important issue, since any effective SBMA therapeutic will likely have to begin at puberty, when normal AR function must be spared to allow for the full expression of male traits.

We removed the sentence as indicated, and we further discussed the relevance of targeting AR toxic GOF without (possibly) exacerbating LOF. Indeed, this aspect is particularly important in SBMA for three reasons:

- i) Any therapeutic strategy is more likely to work if started at puberty, concomitant with or before appearance of symptoms, as stated by this reviewer.**
- ii) AR is on the X chromosome, and patients are male (not because the disease is recessive but because they have high androgen levels in serum). Males are hemizygous for the mutation, and any therapy suppressing mutant AR will enhance LOF, an aspect that cannot be neglected in an X-linked disease affecting males.**
- iii) SBMA patients show signs of androgen insensitivity syndrome; this aspect is important for a therapy that will be administered for the entire life of the patient. Further enhancement of AR LOF may exacerbate sexual dysfunction, metabolic syndrome and diabetes, depression, and skeletal muscle atrophy. Co-factors are critical for proper activity of steroid receptors. Here, we identify two AR co-factors that are aberrantly expressed in muscle. These are certainly not the only ones, and the more we understand about factors that regulate AR activity and that are dysregulated in SBMA (as in cancer), the more likely that we will reach our goal—to develop a**

therapy that can be administered chronically with minimal side effects.

We have clearly stated in the revised manuscript that we cannot rule out the possibility that diminished expression of two key AR co-regulators at least in part enhances mutant AR LOF, even if these co-regulators are overexpressed in skeletal muscle. Please see the first paragraph of the revised Discussion section.

2. While studying a range of models often adds strength to a scientific report, in this case, added strength is much less clear. For example, there is no justification for why some models were chosen over others. Few of the mechanistic studies were conducted using cultured myotubes, despite the clear focus on skeletal muscles in other ways. Why were human stem cells differentiated into motoneurons and not muscle cells? This is particularly important in the context of steroid receptor coregulators that can have, indeed do have, cell specific effects. This is not a trivial issue and should be addressed in the Discussion, including an explanation for why different models were chosen and also acknowledging how studies on one cell type may not provide insight into the mechanisms occurring in another cell type.

We agree that the effect of LSD1 and PRMT6 on AR function is cell type-specific, and in myofibers and motor neurons it is very likely to exert specific functions that result in diverse regulation of gene expression. The reason why we analyzed co-localization and interaction of polyQ-expanded AR with LSD1 and PRMT6 in MN1 and iPSCs differentiated to motor neurons was to show that these AR co-regulators interact with polyQ-expanded AR in a disease-relevant context. We acknowledge that findings in one cell type may not be extended to different cell types, raising the need to investigate pathogenetic pathways in a cell-specific context. We also pointed out that LSD1 and PRMT6 are likely not the only co-regulators dysregulated in SBMA—further investigation may fill this knowledge gap, helping us to better understand how AR works in different tissues in pathophysiological conditions. We added relevant text to the Discussion section (first and last paragraphs).

3. While over-expression of LSD1/PRMT6 is shown to be androgen-dependent in muscle (Fig 1), in line with the androgen-dependent nature of this disease, the in vitro evidence that each factor complexes with AR is androgen-independent (Pg 6), presumably unexpected. This finding is problematic in light of the androgen-dependence of the disease and also how and when coregulators are recruited to the AR (also typically a ligand-dependent event). This important issue should be addressed in the Discussion.

We thank the reviewer for making this point. Interaction of PRMT6/LSD1 and AR occurs in the cytosol as well as the nucleus. We previously showed that AR phosphorylation by AKT prevents androgen binding ¹, and arginine methylation by PRMT6 and phosphorylation by AKT are mutually exclusive post-translational modifications ². It is possible that PRMT6 methylates AR before phosphorylation by AKT to avoid AR degradation by the ubiquitin-proteasome system. Once in the nucleus in a chromatin-bound state, AR may again recruit LSD1 and PRMT6 to carry out a different function (gene expression regulation). Why LSD1 forms a complex with AR and PRMT6 in the cytosol is unknown.

To further explore relevance of the ligand-dependent effect of LSD1 on AR transactivation, which requires the AF-2 surface on AR, we generated an LSD1 mutant with a defective LXXLL motif, which is required for co-regulators to transactivate AR through the AF-2 surface. We found that mutant LSD1-LXXAA fails to transactivate AR (Fig 3c and Supplementary Fig. 5b). Thus, even if the complex is formed in absence and presence of ligand, in the liganded state the effect of LSD1 likely occurs through the AF-2 surface, which acquires a specific structure upon ligand binding. Perhaps this is associated with lysine methylation of AR and PRMT6, an aspect that will be further explored in future studies. We have added a paragraph to the Discussion section. Finally, by performing CHIP analysis of AR on putative AREs in the promoters of *Prmt6* and *Lsd1*, we detected polyQ-expanded AR occupancy at these cis-active elements, suggesting a functional consequence for LSD1 and PRMT6 interaction of mutant AR in the absence of ligand. Please see response to point 6 below.

4. While the authors make a point of including a positive control for their PLA studies, they do not include negative controls controlling for the possibility that the different primary antisera cross-react with the target proteins. Please include these essential negative controls.

We apologize for this oversight. Representative images of negative controls in which we used only one primary antibody (against AR, PRMT6, or LSD1) but not the primary antibody recognizing the interacting partner are shown in Supplementary Figure 3d of the revised manuscript.

5. The authors assert that denervation is a key aspect of SBMA pathology, listing several proteins that are upregulated after denervation. However, what the authors do not mention is that many such proteins are also increased when the system is less active, i.e. expression of such proteins is activity-dependent, as typically shown using α -bungarotoxin or curare to silence muscles. Evidence is poor that SBMA muscles are physically denervated, even in late-stage diseased mice, but there is compelling evidence for functional denervation, as recent published reports demonstrate. Unless anatomical denervation has been shown in the AR100 model, this issue deserves a more nuanced presentation.

We recently showed that our transgenic mice have morphologically normal neuromuscular junctions up to 8 weeks of age but later show some defects in the absence of motor neuron loss³. As stated by this reviewer, there is functional not physical denervation in SBMA. Importantly, the details of why and how this affects neurotransmission are a matter of current investigation. This finding recapitulates what has been observed in other transgenic and knock-in mouse models of SBMA. We have clarified that SBMA is characterized by functional denervation in the revised manuscript.

6. In vivo expression of coregulators was shown to be androgen-dependent, raising the important question of whether the genes for LSD1 and/or PRMT6 have AREs. Such studies would add strength to the paper but at the least, this question should be discussed in the Discussion.

To address this important point, we performed bioinformatic analyses to establish whether there are putative AREs in the promoters and enhancers of

***Lsd1* and *Prmt6* cis-active regulatory regions (Fig. 1h). We found putative AREs in both genes, suggesting that AR may directly regulate their expression. To determine whether AR binds to the ARE of these genes in C2C12 myotube-differentiated cells, we performed chromatin-immunoprecipitation (ChIP) assays in cells expressing AR24Q or AR100Q, as previously described ⁴. Normal AR bound the *Lsd1* promoter, and binding was enhanced upon polyQ expansion. In the case of *Prmt6*, only polyQ-expanded AR bound the ARE. This is an important observation that may explain upregulation of these AR co-regulators in the disease. Our results suggest a feedforward mechanism in the pathological condition, where polyQ-expanded AR enhances expression of its own positive co-regulators, which in turn enhance AR function, leading to dysregulation of gene expression. This aberrant behavior resembles what has been recently described about how AR regulates expression of the transcription co-factor CRY1 to enhance its expression and further enhance AR activity in prostate cancer cells.**

Reviewer #2 (Remarks to the Author):

Review of Nature Communication 3552140

LSD1/PRMT6-targeting gene therapy to attenuate androgen receptor toxic gain-of-function ameliorates SBMA phenotype

SBMA is caused by the expansion of a polyglutamine repeat in the androgen receptor (AR) coding region. The current study is to use Crispr/Cas9 targeting the androgen receptor (AR) co-activators (LSD1/PRMT6) to treat the SBMA. The authors used a variety of cellular and animal models to show that 1) the LSD1 and PRMT6 are co-activators for the AR and that the co-activator expression levels are significantly increased in the presence of the mutant AR (AR100). 2) Crispr/Cas9 knock down the co-activators in the presence of mutant AR can rescue the phenotype in the muscle of transgenic SNMA mice and fly. Since clinical phenotypes of SBMA are featured by motor neuron degeneration and muscle weakness, the authors also used motor neuron derived cell lines and muscle cell lines.

Crispr/Cas9 has been used in a number of studies to test the treatment of rare diseases. The authors also sought to use this method to treat the SBMA animal models. Because targeting the mutant AR will also cause loss of function of the wildtype AR, the authors wanted to target the co-activators of the AR to achieve more selective therapeutic effects. Although the findings are interesting and developing an efficient treatment for the inherited neurodegenerative disease is urgently needed, there are some concerns that need to be addressed by additional experiments or at least clarified by discussion.

The present study has been done mostly in overexpression system, such as transgenic AR100 mice, transfected 293T cells and MN1 cells as well as transgenic flies. The use of over-expression and in vitro system in most of experiments in this study may be difficult to avoid artificial influences of overexpressed mutant AR on the phenotypes. The authors claim that they have AR113Q KI mice, muscle tissues from SBMA patients, and iPSC from SBMA but they did not use these samples for most of their critical experiments.

1). The authors claimed that pharmacology inhibition of AR interaction with co-factors causes loss of function of AR but they did not explain why targeted disruption would not have the same effect?

We apologize for not being clear in the original manuscript. In a previous study carried out in SBMA mice, pharmacologic inhibition of AR interaction with transcription co-factors was shown to have beneficial effects ⁵. This pharmacologic approach is expected to be broad and act on all AR co-regulators, thus resulting in loss-of-function in the long run. Our goal is to target overexpressed co-factors, and not general co-activators or co-repressors expressed at normal levels in vulnerable tissues. This is why we developed an approach that is not designed to silence the target genes but rather to normalize their expression in vivo. This is particularly important for LSD1, as its genetic ablation is lethal ⁶. As stated in the main text, we selected a miRNA that silences *Lsd1* by $\leq 50\%$ in cells. To better clarify this point, we modified the main text in the Discussion section.

2). The authors claimed that the co-activators are in the same complex with AR based on the data of PLA staining. Since they have the specific antibodies, the results would be much stronger to use a co-IP study of SBMA knock-in mice to support this conclusion.

To address this point, we performed pull-down assays in quadriceps muscle samples of 8-week-old WT and AR113Q knock-in mice. Using an anti-PRMT6 antibody for pull-down, we found that PRMT6 forms a complex with AR and LSD1 in vivo. These results are shown in Fig. 4c of the revised manuscript. These results provide in vivo confirmation of our findings from cultured cells.

3). Increased LSD1/PRMT6 mRNA in muscle could result from muscle degeneration. In SBMA, motor neuronal degeneration is the primary cause of muscle degeneration. The current manuscript focused on LSD1/PRMT6 up-regulation in muscle without experiments to explain why these co-factors were upregulated in the SBMA models.

To determine why LSD1 and PRMT6 are overexpressed in SBMA muscle, we performed bioinformatic analysis to search for putative androgen-responsive elements (AREs) in the distal/core promoters and enhancers of these genes. We found a putative ARE in the promoter of both *Lsd1* and *Prmt6* genes. We performed ChIP assays and detected polyQ-expanded AR occupancy at both AREs. Importantly, for the *Lsd1* promoter binding was observed for normal AR and was enhanced for polyQ-expanded AR. For the *Prmt6* promoter, binding was detected specifically for polyQ-expanded AR (Fig. 1h). This important piece of data provides insights into the molecular mechanism underlying overexpression of LSD1 and PRMT6 in SBMA muscle. Please see also response to reviewer #1 point 6.

In figure 1, most data are RT-PCR with the limited Western blotting result in Fig. 1B whose quantification does not seem to be consistent with RT-PCR results.

We reanalyzed the data and now report results of new quantification in a graph that enables quantification of the different samples. We confirm that LSD1 is upregulated by ~9-fold and PRMT6 by ~5-fold in SBMA compared to WT skeletal muscle (Fig. 1b and Supplementary Fig. 1a).

Have the authors examined the expression of LSD1/PRMT6 proteins in the brainstem?

We performed western blotting analysis of spinal cord samples from WT and AR100Q mice at 12 weeks of age and found no difference in LSD1 and PRMT6 expression at the protein level. Data are in Supplementary Fig. 1a.

Minor concerns.

1). Figure 1b western blot should show all the samples

We have now added a panel to show all samples used for quantification of LSD1 and PRMT6 levels in vivo (n = 4 WT and AR100Q mouse skeletal

muscles). These data are presented in Supplementary Fig. 1a of the revised manuscript.

2). Please describe how the transactivation assay was done.

We have added this information to the revised Methods section.

3). Western blots should show molecular weight.

We have added molecular weights to each panel.

Reviewer #3 (Remarks to the Author):

Spinobulbar muscular atrophy (SBMA) is caused by a CAG repeat expansion in the AR gene. PolyQ-expanded AR causes SBMA via gain-of-function (GOF) mechanism but partial loss-of-function (LOF) causes additional phenotypes, making therapeutic interventions complicated. Interaction of AR with its transcriptional co-regulators is necessary for GOF toxicity, however loss of these interactions may enhance the LOF SBMA phenotypes. Therefore, the authors here set out to identify AR co-regulators which are overexpressed in a region-specific manner and found LSD1 and PRMT6 overexpressed in skeletal muscle in patients and mice with SBMA. Pharmacologic and genetic manipulation of these factors and their interactions with AR attenuate SBMA in flies and mice.

The authors do a thorough job of demonstrating overexpression of *Lsd1* and *Prmt6* specifically in the skeletal muscle but not in other tissues in various SBMA mouse models, however, in human data they only show that these genes are overexpressed in skeletal muscle. Further data are needed in non-skeletal human tissue or cells to demonstrate that this phenomenon occurs in humans in addition to mice. The authors convincingly show that LSD1, like PRMT6, is a co-activator of polyQ-expanded AR and that by manipulating these interactions, AR transactivation can be decreased.

Next, the authors test a miRNA knockdown strategy in AR24Q and AR100Q mice, using an AAV9 vector containing miRNAs for both *Lsd1* and *Prmt6*. The data convincingly show that significant knockdown of both LSD1 and PRMT6 lead to improvements in behavioral SBMA readouts, however data are lacking regarding potential negative effects of systemic *Lsd1* and *Prmt6* knockdown, and more transcriptional analyses are needed to explain why just a 10% rescue is seen.

There are a few things that the authors need to do to wrap up this interesting story:

1. Demonstrate in human non-skeletal cells or tissue that *Prmt6* and *Lsd1* are not overexpressed to truly conclude that in humans, like mice, these factors are overexpressed in a tissue-specific manner.

To address this important point, we obtained liver biopsy samples from three SBMA patients and three healthy controls and measured *LSD1* and *PRMT* transcript levels in these samples. We did not find increased expression of these genes in SBMA liver. To further corroborate our findings, we verified that *LSD1* and *PRMT* transcript levels were similar in patient and healthy control iPSC-derived motor neurons. These data are in Supplementary Figure 1b of the revised manuscript.

2. Demonstrate that systemic knockdown of *Lsd1* and *Prmt6* is not toxic. This is the primary concern for the paper, since the premise is that this method corrects toxic GOF without causing any toxic LOF, this needs to be further supported with additional data.

We based the idea that knockdown corrects GOF without causing toxic LOF on the observation that *Lsd1* and *Prmt6* knockdown had no significant effect on the body weight and motor function of WT mice. These data are now presented in the main manuscript to support our conclusion that silencing of *Lsd1* and *Prmt6* is not toxic in WT mice (Figure 8a). These results are

consistent with previous observations that *Prmt6* ablation as well as *Lsd1* haploinsufficiency has no effect on body weight, motor function, and survival in WT mice^{6,7}. Concerning the disease condition, our results show that silencing these AR co-regulators in tissues in which they are overexpressed, namely skeletal muscle (see Fig. 1 and Supplementary Fig. 7 and response to comment 2a below), has beneficial effects on the disease phenotype.

a. Assess more regions such as heart, lungs, adipose tissue, etc to test if PRMT6 and LSD1 protein levels are decreased upon IP injection with AAV9 amir-Lsd1/Prmt6.

To address this point, we performed western blot analysis to detect levels of LSD1 and PRMT6 in spinal cord, liver, heart, epididymal adipose tissue, and lungs of AR100Q mice treated with vehicle or amiR. We found no effect on the levels of AR co-regulators, except for LSD1 in the heart. These results are shown in Supplementary Fig. 7. These results are important, as they show an effect of the amiR in skeletal muscle (Fig. 6 and Supplementary Fig. 7) but not in other tissues. This result is consistent with our model in which polyQ expansions enhance AR toxic gain-of-function, resulting in aberrant transcription of at least two co-regulators in skeletal muscle. These co-regulators then further enhance polyQ-expanded AR transactivation, leading to myofiber degeneration. Perhaps lack of effect in tissues other than skeletal muscle is due to the fact that the pathological processes occurring in these tissues are not associated with LSD1/PRMT6-mediated toxic gain-of-function. We thank the reviewer for suggesting this point, which we believe strengthens our conclusions.

b. Assess expression levels of AR targets in non-skeletal tissue to determine if levels change with PRMT6 and LSD1 reduction.

c. Assess expression levels of PRMT6 complex proteins and their known targets as PRMT6 is a general co-regulator and not only an AR co-activator.

To address these points, we measured transcript levels of *LSD1*, *PRMT6*, and *AR* target genes in the spinal cord, liver, heart, epididymal adipose tissue (white adipose tissue), lungs, and testis of AR100Q mice treated with vehicle and amiR-*Lsd1/Prmt6*. We selected genes, such as *Atp2a1*, *Cdkn1a*, *Fkbp4*, *Fkbp5*, *p53*, and *Ezh2*, that are regulated by LSD1, PRMT6, or AR^{2,7-9}. We found that amiR treatment modified expression of a few genes, indicating that the level of *Lsd1* and *Prmt6* silencing was not sufficient to elicit significant widespread changes in the expression of these targets. Of notice, gene expression changes were observed in the heart, where we found that treatment significantly decreased LSD1 expression (Supplementary Fig. 7). These results are presented in Supplementary Fig. 8 of the revised manuscript.

3. The rescue of 10% of RNA-seq changes is very modest and the comparison of genes rescued in this model to public data sets in other SBMA models is weak.

We agree that gene expression changes associated with treatment were modest, as stated in the main manuscript. We strongly believe that LSD1 and PRMT6 are two out of many epigenetic factors (writers, readers, and erasers) that contribute to the toxic gain-of-function of polyQ-expanded AR. This manuscript represents a proof-of-concept that similar mechanisms (enhanced

expression and function of AR co-regulators) contribute to cancer as well as neurodegeneration, thus offering the opportunity for intervention targeting these co-regulators rather than the receptor itself.

a. Conducting RNA-seq in the WT mice treated with amir-Lsd1/Prmt6 group would be beneficial to determine if the low rescue is due to negative effects of miRNA masking the rescue or it is simply due to modest improvements in the transcriptome.

To address this concern, we performed transcriptomic analysis in WT mice treated with vehicle and amiRs. We did not find any significant change in gene expression associated with treatment. These results are in Supplementary Table 4.

b. It would be nice to see knockdown of these co-regulators in an SBMA human cell model with transcriptional analyses conducted to test if human cells show rescue of SBMA transcriptional changes with Lsd1 and Prmt6 knockdown in addition to mice.

This is a key point to determine whether our miRNA strategy is applicable to patients. To address this point, we designed amiRs targeting human *LSD1* and *PRMT6*. We used the same approach as for mouse amiRs. We selected three amiRs (#1, #2, and #3) and tested them for target silencing in human cells. We found that amiR-*LSD1*#1 and amiR-*PRMT6*#1 significantly silenced the target gene in HEK293T cells, whereas amiR-*LSD1*#1 and #3 and amiR-*PRMT6*#1 and #2 worked well in the androgen-sensitive prostate cancer cell line LNCaP (Fig 9a of the revised manuscript).

Then, we asked whether *LSD1* and *PRMT6* silencing modified expression of endogenous genes. As targets, we selected two genes regulated by AR and its co-regulators: *ATP2A1*, coding for SERCA2; and *CDKN1A*, coding for p21^{CIP1} (Fig. 9b of the revised manuscript)^{2,7,10}. Silencing *LSD1* and *PRMT6* modified expression of the two genes, showing that our human amiRs were effective in cells. In LNCaP cells, we observed significantly reduced proliferation in cells treated with amiRs targeting either *LSD1* or *PRMT6*, showing that these amiRs exert a biological effect in prostate cancer cells (Fig. 9c of the revised manuscript). These data strengthen our conclusions, broaden the impact of our findings, and support further research towards clinical translation of our preclinical strategy not only for SBMA but also possibly for cancer types with a sex bias.

4. Add WT CTR treatment group to weight figure in 8A.

We added the control group in Fig. 8a. Notice that in the hanging wire panel, the line of WT-CTR and amiR-CTR mice overlaps.

5. WT amir-Lsd1/Prmt6 genotype could be moved from supplemental figure 7 to main figure 8A for weight and rotarod so it is clear that the amir-Lsd1/Prmt6 treatment does not cause a deficit independent of disease.

We have modified Fig. 8. We thank the reviewer for this suggestion, which gave us the opportunity to show that silencing of *LSD1* and *PRMT6* is not toxic in WT mice, as this knock-down did not result in any change in body weight,

hanging wire, and rotarod results in the control cohort of mice used in our preclinical study.

6. There is a mistake in supplemental figure 7 – the legend states this figure should be weight and rotarod but the graphs show hanging wire and rotarod.

We apologize for this mistake and have corrected the text.

References

- 1 Palazzolo, I. *et al.* Akt blocks ligand binding and protects against expanded polyglutamine androgen receptor toxicity. *Hum Mol Genet* **16**, 1593-1603, doi:10.1093/hmg/ddm109 (2007).
- 2 Scaramuzzino, C. *et al.* Protein arginine methyltransferase 6 enhances polyglutamine-expanded androgen receptor function and toxicity in spinal and bulbar muscular atrophy. *Neuron* **85**, 88-100, doi:10.1016/j.neuron.2014.12.031 (2015).
- 3 Chivet, M. *et al.* Polyglutamine-Expanded androgen receptor alteration of skeletal muscle homeostasis and myonuclear aggregation are affected by sex, age and muscle metabolism. *Cells* **9**, doi:cells9020325 [pii] 10.3390/cells9020325 (2020).
- 4 Lim, W. F. *et al.* Gene therapy with AR isoform 2 rescues spinal and bulbar muscular atrophy phenotype by modulating AR transcriptional activity. *Sci Adv* **7**, doi:10.1126/sciadv.abi6896 (2021).
- 5 Nedelsky, N. B. *et al.* Native functions of the androgen receptor are essential to pathogenesis in a Drosophila model of spinobulbar muscular atrophy. *Neuron* **67**, 936-952, doi:10.1016/j.neuron.2010.08.034 (2010).
- 6 Wang, J. *et al.* The lysine demethylase LSD1 (KDM1) is required for maintenance of global DNA methylation. *Nat Genet* **41**, 125-129, doi:10.1038/ng.268 (2009).
- 7 Neault, M., Mallette, F. A., Vogel, G., Michaud-Levesque, J. & Richard, S. Ablation of PRMT6 reveals a role as a negative transcriptional regulator of the p53 tumor suppressor. *Nucleic Acids Res* **40**, 9513-9521, doi:10.1093/nar/gks764 (2012).
- 8 Kim, N. H., Kim, S. N., Seo, D. W., Han, J. W. & Kim, Y. K. PRMT6 overexpression upregulates TSP-1 and downregulates MMPs: its implication in motility and invasion. *Biochem Biophys Res Commun* **432**, 60-65, doi:10.1016/j.bbrc.2013.01.085 (2013).
- 9 Yuan, B. *et al.* LSD1 downregulates p21 expression in vascular smooth muscle cells and promotes neointima formation. *Biochem Pharmacol* **198**, 114947, doi:10.1016/j.bcp.2022.114947 (2022).
- 10 Zhu, L. *et al.* LSD1 inhibition suppresses the growth of clear cell renal cell carcinoma via upregulating P21 signaling. *Acta Pharm Sin B* **9**, 324-334, doi:10.1016/j.apsb.2018.10.006 (2019).

REVIEWERS' COMMENTS

Reviewer #1 (Remarks to the Author):

the revisions are acceptable

Reviewer #2 (Remarks to the Author):

Review of NCOMMS-22-06302A

The authors have performed several experiments and substantially revised the manuscript. Most of the changes have obviously strengthened the results and support their conclusions. However, to make the results sound and to fit with the quality of the papers published in Nature Comm, the authors are encouraged to revise a couple of things.

1. Figure 6 g, the quality of western blots (LSD1 and PRMT6) is not acceptable. The quantification numbers on the bottom of the figure are not in agreement with the figure legend.
2. Figure 8d, western blot of Androgen receptor expression is not acceptable. No clear bands are shown in any of the lane. I cannot imagine how the authors were able to quantify the bands. The authors might try the antibody from Abcam ab105225 and use the mouse testis as a control to obtain a better and clean result.

Reviewer #3 (Remarks to the Author):

The authors addressed my concerns.

Point-by-point response to the reviewers' comments

Reviewer #2 (Remarks to the Author):

The authors have performed several experiments and substantially revised the manuscript. Most of the changes have obviously strengthened the results and support their conclusions. However, to make the results sound and to fit with the quality of the papers published in Nature Comm, the authors are encouraged to revise a couple of things.

1. Figure 6 g, the quality of western blots (LSD1 and PRMT6) is not acceptable. The quantification numbers on the bottom of the figure are not in agreement with the figure legend.

We reanalyzed the quantification of both LSD1 and PRMT6 in the AR100Q mice treated with and without miRNA. For LSD1, we have now quantified a total of n = 6 mice/treatment, and we confirmed that treatment significantly decreased LSD1 by 64%. For PRMT6, we selected three mice/treatment based on the quality of the blots, and we confirmed that PRMT6 was significantly decreased by 20%. All the Western blots quantified are shown in Suppl. Fig. 7.

2. Figure 8d, western blot of Androgen receptor expression is not acceptable. No clear bands are shown in any of the lane. I cannot imagine how the authors were able to quantify the bands. The authors might try the antibody from Abcam ab105225 and use the mouse testis as a control to obtain a better and clean result.

We ordered the antibody suggested by this reviewer, but at this point we are still waiting for its delivery. We performed new SDS-PAGE analysis and stained the nitrocellulose membrane with an aliquot of H280 from Santa Cruz, which is no longer under production. We obtained better quality images that show a clear and neat decrease of polyQ-expanded AR aggregation. We removed the quantification of HMW species and total AR from the figure, as the new image unequivocally shows that treatment decreases mutant AR aggregation and accumulation. This finding is consistent with our previous finding that PRMT6 increases polyQ-expanded AR aggregation and is mutually exclusive with phosphorylation at two phosphorylation sites that prevent androgen binding¹.

Reference

- 1 Scaramuzzino, C. *et al.* Protein arginine methyltransferase 6 enhances polyglutamine-expanded androgen receptor function and toxicity in spinal and bulbar muscular atrophy. *Neuron* **85**, 88-100, doi:S0896-6273(14)01140-4 [pii] 10.1016/j.neuron.2014.12.031 (2015).